# A Riemannian Exponential Augmented Lagrangian Method for Computing the Projection Robust Wasserstein Distance

**Bo Jiang**
Ministry of Education Key Laboratory of NSLSCS
School of Mathematical Sciences, Nanjing Normal University
Nanjing 210023, China
jiangbo@njnu.edu.cn

**Ya-Feng Liu** *
State Key Laboratory of Scientific and Engineering Computing
Institute of Computational Mathematics and Scientific/Engineering Computing
Academy of Mathematics and Systems Science, Chinese Academy of Sciences
Beijing 100190, China
yafliu@lsec.cc.ac.cn

## Abstract

Projection robust Wasserstein (PRW) distance is recently proposed to efficiently mitigate the curse of dimensionality in the classical Wasserstein distance. In this paper, by equivalently reformulating the computation of the PRW distance as an optimization problem over the Cartesian product of the Stiefel manifold and the Euclidean space with additional nonlinear inequality constraints, we propose a Riemannian exponential augmented Lagrangian method (REALM) for solving this problem. Compared with the existing Riemannian exponential penalty-based approaches, REALM can potentially avoid too small penalty parameters and exhibit more stable numerical performance. To solve the subproblems in REALM efficiently, we design an inexact Riemannian Barzilai-Borwein method with Sinkhorn iteration (iRBBS), which selects the stepsizes adaptively rather than tuning the stepsizes in efforts as done in the existing methods. We show that iRBBS can return an $\epsilon$-stationary point of the original PRW distance problem within $\mathcal{O}(\epsilon^{-3})$ iterations, which matches the best known iteration complexity result. Extensive numerical results demonstrate that our proposed methods outperform the state-of-the-art solvers for computing the PRW distance.

## 1 Introduction

The optimal transport (OT) problem has found wide applications in machine learning, representation learning, data sciences, and image sciences; see [21, 5, 42, 1, 14] and the references therein for more details. However, its direct application in machine learning may encounter the issue of the curse of dimensionality since the sample complexity of approximating the Wasserstein distance can grow exponentially in dimension [22, 49]. To resolve this issue, by making an important extension to the sliced Wasserstein distance [43], Paty and Cuturi [41], Deshpande et al. [16], and Niles-Weed and Rigollet [40] proposed to project the distributions to a low-dimensional subspace that maximizes the

---

*Corresponding author.

37th Conference on Neural Information Processing Systems (NeurIPS 2023).

Wasserstein distance between the projected distribution, which can reduce the sample complexity and overcome the issue of the curse of dimensionality [40, 16, 35].

In this paper, we focus on the discrete probability measure case. For $\{x_1, \ldots, x_n\} \subset \mathbb{R}^d$ and $\{y_1, \ldots, y_n\} \subset \mathbb{R}^d$, define $M_{ij} = (x_i - y_j)(x_i - y_j)^\mathsf{T}$ for each $(i, j) \in [n] \times [n]$ with $[n] := \{1, \ldots n\}$. Let $\mathbf{1} \in \mathbb{R}^n$ be the all-one vector and $\delta_x$ be the Dirac delta function at $x$. Given $r = (r_1, \ldots, r_n)^\mathsf{T} \in \Delta^n := \{z \in \mathbb{R}^n \mid \mathbf{1}^\mathsf{T} z = 1, z > 0\}$ and $c = (c_1, \ldots, c_n)^\mathsf{T} \in \Delta^n$, define two discrete probability measures $\mu_n = \sum_{i=1}^n r_i \delta_{x_i}$ and $\nu_n = \sum_{i=1}^n c_i \delta_{y_i}$. For $k \in [d]$, the $k$-dimensional projection robust Wasserstein (PRW) distance between $\mu_n$ and $\nu_n$ is defined as [41]

$$\mathcal{P}_k^2(\mu_n, \nu_n) = \max_{U \in \mathcal{U}} \min_{\pi \in \Pi(r,c)} \langle \pi, C(U) \rangle, \tag{1}$$

where $\langle \cdot, \cdot \rangle$ is the standard inner product in $\mathbb{R}^{n \times n}$, $C(U) \in \mathbb{R}^{n \times n}$ with $[C(U)]_{ij} = \langle M_{ij}, UU^\mathsf{T} \rangle$, $\mathcal{U} = \{U \in \mathbb{R}^{d \times k} \mid U^\mathsf{T} U = I_k\}$ is known as the Stiefel manifold with $I_k$ being the $k$-by-$k$ identity matrix, and $\Pi(r, c) = \{\pi \in \mathbb{R}^{n \times n} \mid \pi \mathbf{1} = r, \pi^\mathsf{T} \mathbf{1} = c, \pi \geq 0\}$. Problem (1) is a nonconcave-convex max-min problem over the Stiefel manifold, which makes it very challenging to solve.

**Related works and motivations**. To compute the PRW distance, Paty and Cuturi [41] proposed two algorithms for solving the subspace robust Wasserstein distance, which is a convex relaxation of problem (1) (without the theoretical guarantee on the relaxation gap). An OT or entropy-regularized OT subproblem with dimension $n$ and a full or top $k$ eigendecomposition of a $d$-by-$d$ matrix needs to be solved exactly at each iteration. Very recently, Lin et al. [34] proposed a Riemannian (adaptive) gradient ascent with Sinkhorn (R(A)GAS) algorithm for solving the following entropy-regularized problem with a small regularization parameter $\eta$:

$$\max_{U \in \mathcal{U}} p_\eta(U), \tag{2}$$

where $p_\eta(U) = \min_{\pi \in \Pi(r,c)} \{\langle \pi, C(U) \rangle - \eta H(\pi)\}$, in which $H(\pi) = -\sum_{ij} \pi_{ij} \log \pi_{ij}$ is the entropy function. They showed that R(A)GAS can return an $\epsilon$-stationary point of PRW problem (1) within $\mathcal{O}(\epsilon^{-4})$ iterations if $\eta = \mathcal{O}(\epsilon)$ in (2). However, at each iteration, R(A)GAS needs to solve a regularized OT problem in relatively high precision, which results in a high computational cost. To reduce the complexity of R(A)GAS, Huang et al. [28, 29] proposed a Riemannian (adaptive) block coordinate descent (R(A)BCD) algorithm for solving an equivalent "min" formulation of (2) as

$$\min_{U \in \mathcal{U}, \alpha, \beta \in \mathbb{R}^n} \mathcal{L}_\eta(\mathbf{x}, \mathbf{11}^\mathsf{T}), \tag{3}$$

where $\mathcal{L}_\eta(\cdot, \cdot)$ is defined in (8) further ahead. By choosing $\eta = \mathcal{O}(\epsilon)$ in (3), they showed that the whole iteration complexity of R(A)BCD to attain an $\epsilon$-stationary point of PRW problem (1) reduces to $\mathcal{O}(\epsilon^{-3})$, which significantly improves the complexity of R(A)GAS.

However, there are two main issues of R(A)BCD and R(A)GAS. First, to compute a solution of problem (1) with relatively high quality, $\eta$ in problem (2) or (3) has to be chosen small, which makes the corresponding problem ill-conditioned and may cause numerical instability in solving it. Second, the performance of the above algorithms is sensitive to the stepsizes in updating $U$. Hence, to achieve a better performance, one has to spend some efforts tuning the stepsizes carefully. Resolving these two main issues demands some novel approaches from both theoretical and computational points of view, and this is the motivation and focus of our paper.

**Contributions**. In this paper, by reformulating (1) as an optimization problem defined over the Cartesian product of the Stiefel manifold and the Euclidean space with additional inequality constraints (see problem (6) further ahead), we can resolve the above-mentioned two issues. Our main contributions are summarized as follows. See also Figure 1 for a summary of the related works and the main results of this paper.

(i) We propose a Riemannian exponential augmented Lagrangian method (REALM) to efficiently and faithfully compute the PRW distance, in which a series of subproblems with dynamically decreasing penalty parameters and adaptively updated multiplier matrices are solved approximately. In theory, we establish the global convergence of REALM in the sense that any limit point of the sequence generated by the algorithm is stationary point of the original problem; see Theorem 2.7. Numerically, REALM always outperforms the Riemannian exponential penalty approach since it could avoid too small penalty parameters in many cases.

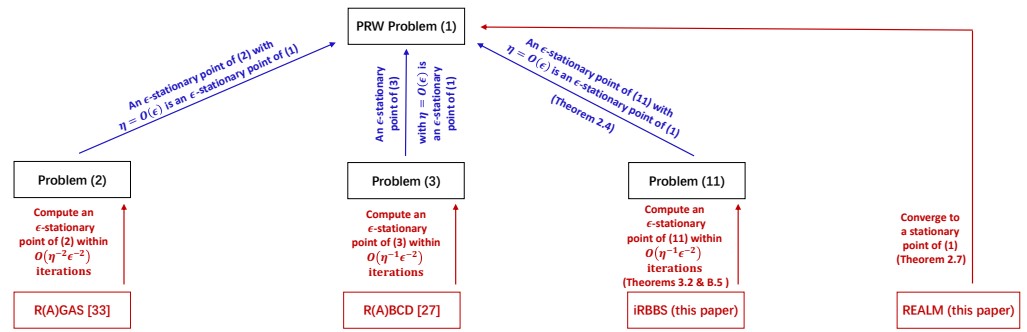

Figure 1: A summary of the related works and the main results of this paper.

(ii) To efficiently solve the subproblem in REALM (i.e., problem (11) further ahead), we view it as a one-block optimization problem over the Stiefel manifold and propose a novel and practical algorithm, namely, the inexact Riemannian Barzilai-Borwein (BB) method with Sinkhorn iteration (iRBBS), wherein a flexible number of Sinkhorn iterations is performed to compute the inexact Riemannian gradient. Compared with R(A)BCD, our proposed iRBBS (applied to problem (11) with fixed $\pi^k$ and $\eta_k = O(\epsilon)$) can not only return a stronger $\epsilon$-stationary point of PRW problem (1) (compared with the definitions in [34, 28]; see Remark 2.2), within $\mathcal{O}(\epsilon^{-3})$ iterations (see Theorem 3.2 with $\epsilon_1 = \epsilon_2 = \epsilon$), but also has a better numerical performance, which mainly benefits from the adaptive Riemannian BB stepsize (based on the inexact Riemannian gradient information).

*Notations.* For $x \in \mathbb{R}^n$, $\mathrm{Diag}(x)$ is an $n \times n$ diagonal matrix with $x$ being its main diagonal. For a matrix $A$, denote $A_{\min} = \min_{ij} A_{ij}$, $A_{\max} = \max_{ij} A_{ij}$, $\|A\|_1 = \sum_{ij} |A_{ij}|$, $\|A\|_\infty = \max_{ij} |A_{ij}|$, $\|A\|_{\mathsf{F}}^2 = \sum_{ij} A_{ij}^2$, and $\|A\|_{\mathrm{var}} = A_{\max} - A_{\min}$. Denote by $\log(\cdot)$ and $\exp(\cdot)$ the element-wise logarithmic and exponential operators, respectively. We use $\mathbb{R}_+^{n \times n}$ and $\mathbb{R}_{++}^{n \times n}$ to denote the nonnegative and positive orthants of $\mathbb{R}^{n \times n}$, respectively. Throughout this paper, we define $C \in \mathbb{R}^{n \times n}$ with $C_{ij} = \|x_i - y_j\|^2$ and $V_\pi = \sum_{ij} \pi_{ij} M_{ij}$.

The tangent space at $U \in \mathcal{U}$ is $\mathrm{T}_U \mathcal{U} = \{\xi \in \mathbb{R}^{d \times k} \mid U^\mathsf{T} \xi + \xi^\mathsf{T} U = 0\}$. Let $\mathrm{T}\mathcal{U} = \{(U, \xi) \mid U \in \mathcal{U} \text{ and } \xi \in \mathrm{T}_U \mathcal{U}\}$ be the tangent bundle of $\mathcal{U}$. A smooth map $\mathsf{Retr} : \mathrm{T}\mathcal{U} \to \mathcal{U} : (U, \xi) \mapsto \mathsf{Retr}_U(\xi)$ is called a retraction if each curve $\mathcal{R}(t) = \mathsf{Retr}_U(t\xi)$ satisfies $\mathcal{R}(0) = U$ and $\mathcal{R}'(0) = \xi$; see [9, Definition 3.47] or [2, Definition 4.1.1]. The Riemannian metric $\langle \cdot, \cdot \rangle_U$ endowed on the Stiefel manifold $\mathcal{U}$ is taken as the standard metric $\langle \cdot, \cdot \rangle$ on $\mathbb{R}^{d \times k}$. The Riemannian gradient of a smooth function $f : \mathbb{R}^{d \times k} \to \mathbb{R}$ at $U \in \mathcal{U}$ is defined as $\mathrm{grad}\, f(U)$, which satisfies $\langle \mathrm{grad}\, f(U), \xi \rangle_U = \langle \nabla f(U), \xi \rangle$ for all $\xi \in \mathrm{T}_U \mathcal{U}$, where $\nabla f(U)$ denotes the Euclidean gradient of $f$ at $U$. If $U^\mathsf{T} \nabla f(U)$ is symmetric, we have $\mathrm{grad}\, f(U) = \mathsf{Proj}_{\mathrm{T}_U \mathcal{U}}(\nabla f(U)) = (I_d - UU^\mathsf{T})\nabla f(U)$.

The rest of this paper is organized as follows. The proposed REALM is introduced in Section 2. A practical iRBBS for solving the subproblem in REALM is proposed in Section 3. Numerical results are presented in Section 4. Finally, we draw some concluding remarks in Section 5.

## 2 A Riemannian Exponential ALM for Computing the PRW Distance (1)

Given a fixed $U \in \mathcal{U}$, consider the OT problem

$$\min_{\pi \in \mathbb{R}^{n \times n}} \langle \pi, C(U) \rangle \quad \text{s.t.} \quad \pi \mathbf{1} = r, \ \pi^\mathsf{T} \mathbf{1} = c, \ \pi \geq 0. \tag{4}$$

By adding a redundant constraint $\|\pi\|_1 = 1$ [29, 36], we derive the dual of (4) as

$$\max_{\alpha \in \mathbb{R}^n, \beta \in \mathbb{R}^n} -(r^\mathsf{T}\alpha + c^\mathsf{T}\beta + y) \quad \text{s.t.} \quad \varphi(\mathbf{x})_{ij} + y \geq 0, \quad \forall (i, j) \in [n] \times [n], \tag{5}$$

where $\mathbf{x} = (\alpha, \beta, U)$ and $\varphi(\mathbf{x}) \in \mathbb{R}^{n \times n}$ with $\varphi(\mathbf{x})_{ij} = \alpha_i + \beta_j + \langle M_{ij}, UU^\mathsf{T} \rangle$ for each $(i, j) \in [n] \times [n]$. Note that the matrix $\pi$ in (4) can also be understood as the Lagrange multiplier corresponding to the inequalities in (5). Therefore, the value $\mathcal{P}_k^2(\mu_n, \nu_n)$ defined in (1) is equal to the opposite of the optimal objective value of the following optimization problem:

$$\min_{\mathbf{x} \in \mathcal{M}, y \in \mathbb{R}} r^\mathsf{T}\alpha + c^\mathsf{T}\beta + y \quad \text{s.t.} \quad \varphi(\mathbf{x})_{ij} + y \geq 0, \quad \forall (i, j) \in [n] \times [n], \tag{6}$$

where $\mathcal{M} = \mathbb{R}^n \times \mathbb{R}^n \times \mathcal{U}$. Motivated by the first-order necessary condition of problem (6) (see Appendix A.1 for details), we define the $(\epsilon_1, \epsilon_2)$-stationary point of problem (1) as follows.

**Definition 2.1.** We call $(\tilde{\mathbf{x}}, \tilde{\pi}) \in \mathcal{M} \times \Pi(r, c)$ an $(\epsilon_1, \epsilon_2)$-stationary point of PRW problem (1), if $\|\mathsf{Proj}_{\mathrm{T}_{\tilde{U}}\mathcal{U}}(-2V_{\tilde{\pi}}\tilde{U})\|_{\mathsf{F}} \leq \epsilon_1$ and $\langle \tilde{\pi}, Z(\tilde{\mathbf{x}}) \rangle \leq \epsilon_2$, where $Z(\tilde{\mathbf{x}}) \in \mathbb{R}^{n \times n}$ with $Z(\tilde{\mathbf{x}})_{ij} = \varphi(\mathbf{x})_{ij} - \varphi(\mathbf{x})_{\min}$. If $\epsilon_1 = \epsilon_2 = 0$, we call such $(\tilde{\mathbf{x}}, \tilde{\pi})$ a stationary point of PRW problem (1).

*Remark* 2.2. Our Definition 2.1 is stronger than [28, Definition 3.1] and [34, Definition 2.7] in the sense that the $(\epsilon_1, \epsilon_2)$-stationary point satisfying the conditions here also satisfies all conditions therein. See Appendix A.2 for more details.

Given $\pi \in \mathbb{R}_{++}^{n \times n}$ and $\eta > 0$, define the function $\zeta_\eta(\mathbf{x}, \pi) \in \mathbb{R}^{n \times n}$ with

$$[\zeta_\eta(\mathbf{x}, \pi)]_{ij} = \pi_{ij} \exp\left(-\frac{\varphi(\mathbf{x})_{ij}}{\eta}\right), \tag{7}$$

define

$$\widetilde{\mathcal{L}}_\eta(\mathbf{x}, y, \pi) = r^{\mathsf{T}}\alpha + c^{\mathsf{T}}\beta + y + \eta \sum_{ij} \pi_{ij} \exp\left(-\frac{\varphi(\mathbf{x})_{ij} + y}{\eta}\right)$$

and

$$\mathcal{L}_\eta(\mathbf{x}, \pi) := r^{\mathsf{T}}\alpha + c^{\mathsf{T}}\beta + \eta \log(\|\zeta_\eta(\mathbf{x}, \pi)\|_1). \tag{8}$$

One natural approach for solving problem (6) is the Riemannian exponential penalty approach (where the manifold constraints are kept in the subproblem), which aims to solve the penalty subproblem

$$\min_{\mathbf{x} \in \mathcal{M}, y \in \mathbb{R}} \widetilde{\mathcal{L}}_\eta(\mathbf{x}, y, \mathbf{1}\mathbf{1}^{\mathsf{T}}). \tag{9}$$

For any fixed $x$, letting $\nabla_y \widetilde{\mathcal{L}}_\eta(\mathbf{x}, y, \mathbf{1}\mathbf{1}^{\mathsf{T}}) = 0$, we can obtain the optimal $y$ as $y = \eta \log(\|\zeta_\eta(\mathbf{x}, \mathbf{1}\mathbf{1}^T)\|_1)$. By eliminating the variable $y$ in (9), we thus obtain the subproblem (3) of the approach in Huang et al. [28].

It is known that the exponential augmented Lagrangian method (ALM) is usually more stable than the exponential penalty approach; see [18, Tables 3.1-3.3] for a detailed example. More specifically, the penalty parameter in the exponential ALM can be chosen as any positive number in the convex case [47, 52] or can be bounded away from zero under some standard assumptions in the general nonlinear case [20], which is in sharp contrast to the exponential penalty approach. Based on the aforementioned knowledge, we thus extend the exponential ALM [8] to the manifold case to solve problem (6). Fix the current estimate of the Lagrange multiplier corresponding to the inequality constraints in (6) and the penalty parameter as $\pi^k$ and $\eta_k$, respectively. Then the subproblem at the $k$-th iteration is given as

$$\min_{\mathbf{x} \in \mathcal{M}, y \in \mathbb{R}} \widetilde{\mathcal{L}}_{\eta_k}(\mathbf{x}, y, \pi^k). \tag{10}$$

Similar to the way for eliminating $y$ in (9), we obtain an equivalent formulation of (10):

$$\min_{\mathbf{x} \in \mathcal{M}} \mathcal{L}_{\eta_k}(\mathbf{x}, \pi^k). \tag{11}$$

Define the matrix $\phi_{\eta_k}(\mathbf{x}, \pi^k) \in \mathbb{R}^{n \times n}$ with

$$[\phi_{\eta_k}(\mathbf{x}, \pi^k)]_{ij} = [\zeta_{\eta_k}(\mathbf{x}, \pi^k)]_{ij} / \|\zeta_{\eta_k}(\mathbf{x}, \pi^k)\|_1. \tag{12}$$

By the chain rule, we have $\nabla_\alpha \mathcal{L}_{\eta_k}(\mathbf{x}, \pi^k) = r - \phi_{\eta_k}(\mathbf{x}, \pi^k)\mathbf{1}$, $\nabla_\beta \mathcal{L}_{\eta_k}(\mathbf{x}, \pi^k) = c - \phi_{\eta_k}(\mathbf{x}, \pi^k)^{\mathsf{T}}\mathbf{1}$, $\nabla_U \mathcal{L}_{\eta_k}(\mathbf{x}, \pi^k) = -2V_{\phi_{\eta_k}(\mathbf{x}, \pi^k)}U$, and $\mathrm{grad}_U \mathcal{L}_{\eta_k}(\mathbf{x}, \pi^k) = \mathsf{Proj}_{\mathrm{T}_U\mathcal{U}}(-2V_{\phi_{\eta_k}(\mathbf{x}, \pi^k)}U)$. Let $\mathsf{e}_{\eta_k}^1(\mathbf{x}, \pi^k) = \|\mathrm{grad}_U \mathcal{L}_{\eta_k}(\mathbf{x}, \pi^k)\|_{\mathsf{F}}$ and $\mathsf{e}_{\eta_k}^2(\mathbf{x}, \pi^k) = \|\nabla_\alpha \mathcal{L}_{\eta_k}(\mathbf{x}, \pi^k)\|_1 + \|\nabla_\beta \mathcal{L}_{\eta_k}(\mathbf{x}, \pi^k)\|_1$. The $(\epsilon_1, \epsilon_2)$-stationary point of (11) and the connections of the approximate stationary points of problems (11) and (1) are given as follows.

**Definition 2.3.** We say $\tilde{\mathbf{x}} \in \mathcal{M}$ an $(\epsilon_1, \epsilon_2)$-stationary point of problem (11) (with fixed $\eta_k$ and $\pi^k$) if $\mathsf{e}_{\eta_k}^1(\tilde{\mathbf{x}}, \pi^k) \leq \epsilon_1$ and $\mathsf{e}_{\eta_k}^2(\tilde{\mathbf{x}}, \pi^k) \leq \epsilon_2$.

**Theorem 2.4.** *Suppose* $\tilde{\mathbf{x}} = (\tilde{\alpha}, \tilde{\beta}, \tilde{U}) \in \mathcal{M}$ *with* $\|\zeta_{\eta_k}(\tilde{\mathbf{x}}, \pi^k)\|_1 = 1$ *is an* $(\epsilon_1, \epsilon_2)$-*stationary point of problem* (11). *Then, we have*

$$\|\mathsf{Proj}_{\mathrm{T}_{\tilde{U}}\mathcal{U}}(-2V_{\hat{\pi}}\tilde{U})\|_{\mathsf{F}} \leq \epsilon_1 + 2\|C\|_\infty \epsilon_2, \tag{13a}$$

$$\langle \hat{\pi}, Z(\tilde{\mathbf{x}}) \rangle \leq (2\log n + \|\log \pi^k\|_{\mathrm{var}})\eta_k + (\|\tilde{\alpha}\|_{\mathrm{var}} + \|\tilde{\beta}\|_{\mathrm{var}} + \|C\|_\infty)\epsilon_2, \tag{13b}$$

*where* $\hat{\pi} := \mathtt{Round}(\phi_{\eta_k}(\tilde{\mathbf{x}}, \pi^k), \Pi(r, c))$ *is a feasible matrix returned by running the rounding procedure "*$\mathtt{Round}$*" given in [3, Algorithm 2] with input* $\phi_{\eta_k}(\tilde{\mathbf{x}}, \pi^k)$.

*Remark* 2.5. By Theorem 2.4, we can see that, given any $\epsilon > 0$, for fixed $\pi^k$, by choosing $\eta_k = O(\epsilon)$ with $\epsilon_1 = \epsilon_2 = \epsilon$, an $\epsilon$-stationary point with bounded $(\tilde{\alpha}, \tilde{\beta})$ (which can be found efficiently by iRBBS proposed in Section 3) of problem (11) can recover an $\epsilon$-stationary point of PRW problem (1). This may also be of independent interest for the case with fixed $U$. In such case, one can return a feasible $\epsilon$-stationary point of the OT problem (4) (with fixed $U$) evaluated by the primal-dual gap other than the primal gap typically used in the literature, such as [3, Theorem 1].

Let $\mathbf{x}^0$ be an initial point satisfying $r^\mathsf{T}\alpha^0 = c^\mathsf{T}\beta^0$ and $\|\zeta_{\eta_1}(\mathbf{x}^0, \pi^1)\|_1 = 1$ with $\pi^1 = \mathbf{1}\mathbf{1}^\mathsf{T}$. We require that $\mathbf{x}^k = (\alpha^k, \beta^k, U^k)$, the $(\epsilon_{k,1}, \epsilon_{k,2})$-stationary point of the subproblem (11) with $k \geq 1$, satisfies the following conditions:

$$r^\mathsf{T}\alpha^k = c^\mathsf{T}\beta^k, \ \|\zeta_{\eta_k}(\mathbf{x}^k, \pi^k)\|_1 = 1, \ \mathcal{L}_{\eta_k}(\mathbf{x}^k, \pi^k) \leq \min\{\mathcal{L}_{\eta_k}(\mathbf{x}^{k-1}, \pi^k), \mathcal{L}_{\eta_k}(\mathbf{x}^0, \pi^k)\}. \quad (14)$$

These conditions are important to establish the convergence of REALM (as shown in Appendix A.4). The following key observation to (11) shows that $\mathbf{x}^k$ satisfying the first two conditions in (14) can be easily obtained. We omit the detailed proof for brevity since they can be verified easily by noticing that $\mathcal{L}_{\eta_k}(\mathbf{x}, \pi^k) = \mathcal{L}_{\eta_k}(\mathbf{x}^s, \pi^k)$ where $\mathbf{x}^s$ is defined in Proposition 2.6. The third condition in (14) can be easily satisfied by using any descent-like method starting from the better point of $\mathbf{x}^{k-1}$ and $\mathbf{x}^0$ to solve problem (11).

**Proposition 2.6.** *For any* $\mathbf{x} = (\alpha, \beta, U) \in \mathcal{M}$, *consider* $\mathbf{x}^s = (\alpha + \upsilon_1\mathbf{1}, \beta + \upsilon_2\mathbf{1}, U) \in \mathcal{M}$ *with* $\upsilon_1 = (c^\mathsf{T}\beta - r^\mathsf{T}\alpha + \eta_k \log(\|\zeta_{\eta_k}(\mathbf{x}, \pi^k)\|_1))/2$ *and* $\upsilon_2 = (r^\mathsf{T}\alpha - c^\mathsf{T}\beta + \eta_k \log(\|\zeta_{\eta_k}(\mathbf{x}, \pi^k)\|_1))/2$. *Then we have that* $r^\mathsf{T}(\alpha + \upsilon_1\mathbf{1}) = c^\mathsf{T}(\beta + \upsilon_2\mathbf{1})$ *and* $\|\zeta_{\eta_k}(\mathbf{x}^s, \pi^k)\|_1 = 1$ *and also that* $\mathcal{L}_{\eta_k}(\mathbf{x}, \pi^k) = \mathcal{L}_{\eta_k}(\mathbf{x}^s, \pi^k)$ *and* $\nabla_{\mathbf{x}}\mathcal{L}_{\eta_k}(\mathbf{x}, \pi^k) = \nabla_{\mathbf{x}}\mathcal{L}_{\eta_k}(\mathbf{x}^s, \pi^k)$.

With such $\mathbf{x}^k$ in hand, we compute the candidate of the next estimate $\pi^{k+1}$ as

$$\tilde{\pi}^{k+1} = \phi_{\eta_k}(\mathbf{x}^k, \pi^k). \quad (15)$$

Denote $W^k \in \mathbb{R}^{n \times n}$ with $W^k_{ij} = \min\{\eta_k\tilde{\pi}^{k+1}_{ij}, \varphi(\mathbf{x}^k)_{ij}\}$. The penalty parameter $\eta_{k+1}$ is updated according to the progress of the complementarity violation [4], denoted by $\|W^k\|_\mathsf{F}$. If $\|W^k\|_\mathsf{F} \leq \gamma_W\|W^{k-1}\|_\mathsf{F}$ with $\gamma_W \in (0, 1)$, we keep $\eta_{k+1} = \eta_k$ and update $\pi^{k+1} = \tilde{\pi}^{k+1}$; otherwise we keep $\pi^{k+1} = \pi^k$ and reduce $\eta_{k+1}$ via

$$\eta_{k+1} = \min\left\{\gamma_\eta\eta_k, \varrho_k/\|\log\pi^k\|_\infty\right\}, \quad (16)$$

where $\gamma_\eta \in (0, 1)$ is a constant and $\varrho_k \to 0$ with $\varrho_k > 0$.

We summarize the above discussion as the complete algorithm in Algorithm 1, whose convergence is established as follows.

**Theorem 2.7.** *Let* $\{(\mathbf{x}^k, \tilde{\pi}^k)\}$ *be the sequence generated by Algorithm 1 with* $\epsilon_1 = \epsilon_2 = \epsilon_c = 0$ *and* $(\mathbf{x}^\infty, \tilde{\pi}^\infty)$ *be a limit point of* $\{(\mathbf{x}^k, \tilde{\pi}^k)\}$. *Then,* $(\mathbf{x}^\infty, \tilde{\pi}^\infty)$ *is a stationary point of PRW problem* (1).

*Remark* 2.8. Our proposed REALM is a nontrivial extension of the exponential ALM from the Euclidean case [47, 20, 51, 52] to the Riemannian case. The following two differences distinguish our proposed REALM from the existing exponential ALM (e.g., the one proposed in [20]): (i) **Measure of complementarity.** The measure of complementarity used in our proposed REALM is motivated by the direct use of the complementarity condition adopted in the classical (quadratic) ALM, while that used in [20] is a variant of the measure for the exponential case. (ii) **Conditions on global convergence.** To guarantee the global convergence of the exponential ALMs, some (strong) constraint qualifications, the boundness of the iterates, and the feasibility of the limit point of the iterates generally need to be assumed; see Proposition 2.1 and Theorem 2.1 in [20] for the corresponding results. In contrast, for our considered PRW distance problem (6), we can prove the boundness of the iterates generated by REALM without making the assumption and establish the global convergence of REALM without explicitly dealing with the constraint qualification assumption. This advantage is mainly due to the *essential* changes in the proposed REALM (compared with the existing exponential ALMs), i.e., specific conditions (14) and (16) (motivated by (13b)) on the solution of subproblems and the adopted measure of complementarity.

Moreover, it would be possible to extend the analysis in [20] to prove that the penalty parameter $\eta_k$ in Algorithm 1 is bounded away from zero if the Riemannian versions of the three conditions hold, including the linear independence constraint qualification, the strict complementarity condition, and the second-order sufficient condition. However, these three conditions might not be easy to check since we do not have prior knowledge of the solution.

---
**Algorithm 1:** REALM for solving problem (6).
---
1 **Input:** Choose $\epsilon_c, \epsilon_1, \epsilon_2, \eta_1 > 0$, $\gamma_W, \gamma_\eta, \gamma_\epsilon \in (0,1)$, $\epsilon_{1,1} \geq \epsilon_1, \epsilon_{1,2} \geq \epsilon_2$. Choose $\pi^1 = \mathbf{1}\mathbf{1}^\mathsf{T}$,
 $\mathbf{x}^0 \in \mathcal{M}$ with $r^\mathsf{T}\alpha^0 = c^\mathsf{T}\beta^0$, $\|\zeta_{\eta_1}(\mathbf{x}^0, \pi^1)\|_1 = 1$. Compute $W^0$ with $W_{ij}^0 = \min\{\eta_1, \varphi(\mathbf{x}^0)_{ij}\}$.
2 **for** $k = 1, 2, \ldots,$ **do**
3  Compute an $(\epsilon_{k,1}, \epsilon_{k,2})$-stationary point $\mathbf{x}^k$ of (11) satisfying (14);
4  Compute $\tilde{\pi}^{k+1}$ according to (15) and compute $W^k$;
5  **if** $\|W^k\|_\mathsf{F} \leq \epsilon_c$, $\mathsf{e}_{\eta_k}^1(\mathbf{x}^k, \pi^k) \leq \epsilon_1$, $\mathsf{e}_{\eta_k}^2(\mathbf{x}^k, \pi^k) \leq \epsilon_2$ **then return** $\mathbf{x}^k$ and $\pi^{k+1} = \tilde{\pi}^{k+1}$;
6  **if** $\|W^k\|_\mathsf{F} \leq \gamma_W \|W^{k-1}\|_\mathsf{F}$ **then** set $\pi^{k+1} = \tilde{\pi}^{k+1}$, $\eta_{k+1} = \eta_k$ **else** set $\pi^{k+1} = \pi^k$ and
    update $\eta_{k+1}$ via (16) ;
7  Set $\epsilon_{k+1,1} = \gamma_\epsilon \epsilon_{k,1}$ and $\epsilon_{k+1,2} = \gamma_\epsilon \epsilon_{k,2}$.
---

*Remark* 2.9. We cannot establish the iteration complexity of Algorithm 1 due to the following two main difficulties: (i) characterizing the connection between the two complementarity measures $\|W^k\|_\mathsf{F}$ and $e^k := \langle \hat{\pi}^k, Z(\mathbf{x}^k) \rangle$ with $\hat{\pi}^k := \mathtt{Round}(\phi_{\eta_k}(\mathbf{x}^k, \pi^k), \Pi(r, c))$ at the approximate stationary point of the subproblem; (ii) establishing the relationship between $\eta_k \tilde{\pi}_{ij}^{k+1}$ and $\varphi(\mathbf{x}^k)_{ij}$. Thanks to Theorem 2.4, we can slightly modify Algorithm 1 to establish the iteration complexity. By modifying the "if" condition in Line 6 of Algorithm 1 as "$\|W^k\|_\mathsf{F} \leq \gamma_W \|W^{k-1}\|_\mathsf{F}$ and $e^k \leq \gamma_W e^{k-1}$", and leveraging the connection between the approximate stationary points of the subproblem and the original problem as proven in (13), we know that the modified Algorithm 1 will terminate within at most $\mathcal{O}(\max\{\log \epsilon_1^{-1}, \log \epsilon_2^{-1}, T_k\})$ iterations, where $T_k := \min\{k \mid \varrho_k \leq \epsilon_c\}$.

*Remark* 2.10. Problem (6) can also be solved by the Riemannian ALMs based on the quadratic penalty function [37, 55]. However, the subproblems therein have four blocks of the variable, i.e., $(\alpha, \beta, y, U)$, and some customized solvers are needed to solve them. Moreover, the connections between the stationary points of problem (1) and the subproblems therein remain unclear.

## 3  A Practical iRBBS for Solving Subproblem (11)

At first glance, (11) is a three-block optimization problem and can be solved by R(A)BCD proposed by [28]. However, as stated therein, tuning the stepsizes for updating $U$ is not easy for R(A)BCD. In sharp contrast, we understand (11) as optimization with only one variable $U$ as follows:

$$\min_{U \in \mathcal{U}} \left\{ q(U) := \min_{\alpha \in \mathbb{R}^n, \beta \in \mathbb{R}^n} \mathcal{L}_{\eta_k}(\mathbf{x}, \pi^k) \right\}. \tag{17}$$

By [34, Lemma 3.1], we know that $q(\cdot)$ is differentiable over $\mathbb{R}^{d \times k}$. Here, we give a new formulation of $\operatorname{grad} q(U)$, which can provide more insights into approximating $\operatorname{grad} q(U)$.

**Lemma 3.1.** *Let $(\alpha_U^*, \beta_U^*) \in \operatorname{argmin}_{\alpha \in \mathbb{R}^n, \beta \in \mathbb{R}^n} \mathcal{L}(\pi^k)$ and $\mathbf{x}_U^* = (\alpha_U^*, \beta_U^*, U)$. Then we have* $\operatorname{grad} q(U) = \operatorname{grad}_U \mathcal{L}_{\eta_k}(\mathbf{x}_U^*, \pi^k) = \mathsf{Proj}_{\mathrm{T}_U \mathcal{U}}(-2V_{\phi_{\eta_k}(\mathbf{x}_U^*, \pi^k)} U)$.

Hence we could use the Riemannian gradient descent (RGD) method [2] to solve problem (11). Letting $\tau_t > 0$ be some stepsize, the main iterations of RGD are given as

$$U^{t+1} = \mathsf{Retr}_{U^t}\left( -\tau_t \operatorname{grad} q(U^t) \right). \tag{18}$$

However, RGD (18) needs to calculate $(\alpha_U^*, \beta_U^*)$ exactly, which can be challenging (or might be unnecessary) to do. Motivated by the well-established inexact gradient type methods for optimization in the Euclidean space [11, 26, 46, 17, 39, 7, 52], we propose an inexact RGD framework. Let $\mathbf{x}^t = (\alpha^t, \beta^t, U^t)$ with $(\alpha^t, \beta^t) \approx (\alpha_{U^t}^*, \beta_{U^t}^*)$, wherein the inexactness level is determined by $\mathsf{e}_{\eta_k}^2(\mathbf{x}^t, \pi^k) = \|\nabla_\alpha \mathcal{L}_{\eta_k}(\mathbf{x}^t, \pi^k)\|_1 + \|\nabla_\beta \mathcal{L}_{\eta_k}(\mathbf{x}^t, \pi^k)\|_1 \leq \theta_t$ for given $\theta_t$. By Lemma 3.1, we use

$$\xi^t := \operatorname{grad}_U \mathcal{L}_{\eta_k}(\mathbf{x}^t, \pi^k) = \mathsf{Proj}_{\mathrm{T}_{U^t}\mathcal{U}}\left( -2V_{\phi_{\eta_k}(\mathbf{x}^t, \pi^k)} U^t \right)$$

to approximate $\operatorname{grad} q(U^t)$. Then, we perform an inexact RGD with $\operatorname{grad} q(U^t)$ in (18) replaced by $\xi^t$. More specifically, given the inexactness parameter $\theta_{t+1} \geq 0$ and the stepsize $\tau_t \geq 0$, we update $\mathbf{x}^{t+1} = (\alpha^{t+1}, \beta^{t+1}, U^{t+1})$ with $U^{t+1}$ and $(\alpha^{t+1}, \beta^{t+1}) \approx (\alpha_{U^{t+1}}^*, \beta_{U^{t+1}}^*)$ satisfying

$$U^{t+1} = \mathsf{Retr}_{U^t}\left( -\tau_t \xi^t \right), \qquad \text{(inexact RGD step)} \tag{19a}$$

$$\mathsf{e}_{\eta_k}^2(\mathbf{x}^{t+1}, \pi^k) \leq \theta_{t+1}. \qquad \text{(inexactness criterion)} \tag{19b}$$

To make iRGD (19) practical, the first main ingredient is how to compute $(\alpha^{t+1}, \beta^{t+1})$ such that (19b) holds. Given $U^{t+1} \in \mathcal{U}$, $\alpha^{(0)} = \alpha^t$, and $\beta^{(0)} = \beta^t$, for $\ell = 0, 1, \ldots$, we adopt the block coordinate descent method to update

$$\alpha^{(\ell+1)} = \underset{\alpha \in \mathbb{R}^n}{\operatorname{argmin}} \, \mathcal{L}_{\eta_k}(\alpha, \beta^{(\ell)}, U^{t+1}, \pi^k), \quad \beta^{(\ell+1)} = \underset{\beta \in \mathbb{R}^n}{\operatorname{argmin}} \, \mathcal{L}_{\eta_k}(\alpha^{(\ell+1)}, \beta, U^{t+1}, \pi^k). \quad (20)$$

Note that $\alpha^{(\ell+1)}$ and $\beta^{(\ell+1)}$ admit the closed-form solutions as follows:

$$\alpha^{(\ell+1)} = \alpha^{(\ell)} - \eta_k \log r + \eta_k \log(\zeta^{(\ell)} \mathbf{1}), \quad \beta^{(\ell+1)} = \beta^{(\ell)} - \eta_k \log c + \eta_k \log((\zeta^{(\ell+\frac{1}{2})})^{\mathsf{T}} \mathbf{1}), \quad (21)$$

where $\zeta^{(\ell)} := \zeta_{\eta_k}(\alpha^{(\ell)}, \beta^{(\ell)}, U^{t+1}, \pi^k)$ and $\zeta^{(\ell+\frac{1}{2})} := \zeta_{\eta_k}(\alpha^{(\ell+1)}, \beta^{(\ell)}, U^{t+1}, \pi^k)$ (see (7) for its definition). Note that for fixed $U^{t+1}$ and $\pi^k = \mathbf{11}^{\mathsf{T}}$, (21) reduces to the famous Sinkhorn iteration [15]. Therefore, we still name (21) as the Sinkhorn iteration. It is easy to verify that $\|\zeta^{(\ell+\frac{1}{2})}\|_1 = \|\zeta^{(\ell+1)}\|_1 = 1$ (see also [28, Remark 3.1]). By (12), we have

$$\pi^{(\ell+\frac{1}{2})} := \phi_{\eta_k}(\alpha^{(\ell+1)}, \beta^{(\ell)}, U^{t+1}, \pi^k) = \zeta^{(\ell+\frac{1}{2})}, \quad \pi^{(\ell+1)} := \phi_{\eta_k}(\alpha^{(\ell+1)}, \beta^{(\ell+1)}, U^{t+1}, \pi^k) = \zeta^{(\ell+1)}.$$
$$(22)$$

From the update of $\beta^{(\ell+1)}$, we have $(\pi^{(\ell+1)})^{\mathsf{T}} \mathbf{1} - c = 0$. Therefore, to make condition (19b) hold, we stop the Sinkhorn iteration once

$$\|\pi^{(\ell+1)} \mathbf{1} - r\|_1 \leq \theta_{t+1}, \quad (23)$$

and set $\alpha^{t+1} = \alpha^{(\ell+1)}$, $\beta^{t+1} = \beta^{(\ell+1)}$. Recalling the calculation of the gradient and the definition of $\mathsf{e}_{\eta_k}^2(\mathbf{x}^{t+1}, \pi^k)$ after (12), we have

$$\mathsf{e}_{\eta_k}^2(\mathbf{x}^{t+1}, \pi^k) = \|\pi^{(\ell+1)} \mathbf{1} - r\|_1.$$

Next, we choose the stepsize $\tau_t$ in (19a). Since the accurate function and gradient information of $q(U^t) = \mathcal{L}_{\eta_k}(\mathbf{x}_{U^t}^*, \pi^k)$ is unavailable, we cannot expect to build the linesearch condition based on $q(U^t)$ and need to find some appropriate potential function instead. Considering that $\mathcal{L}_{\eta_k}(\mathbf{x}^t, \pi^k)$ is an approximation of $q(U^t)$ and the approximation error is controlled by $\mathsf{e}_{\eta_k}^2(\mathbf{x}^t, \pi^k)$, it is thus desirable that some combinations of $\mathcal{L}_{\eta_k}(\mathbf{x}^t, \pi^k)$ and $\mathsf{e}_{\eta_k}^2(\mathbf{x}^t, \pi^k)$ will be smaller than the corresponding values at the previous iteration. Given $\rho \in [0, \eta_k/2)$, we define the potential function as

$$E_\rho(\mathbf{x}^t) = \mathcal{L}_{\eta_k}(\mathbf{x}^t, \pi^k) + \rho(\mathsf{e}_{\eta_k}^2(\mathbf{x}^t, \pi^k))^2 \quad (24)$$

and require the stepsize $\tau_t$ to satisfy the following nonmonotone line search condition:

$$E_\rho(\mathbf{x}^{t+1}) \leq E_t^r - \delta_1 \tau_t \|\xi^t\|_{\mathsf{F}}^2 - \left(\frac{\eta_k}{2} - \rho\right)(\mathsf{e}_{\eta_k}^2(\mathbf{x}^{t+1}, \pi^k))^2, \quad (25)$$

where $E_{t+1}^r = (\gamma Q_t E_t^r + E_\rho(\mathbf{x}^{t+1}))/Q_{t+1}$ and $Q_{t+1} = \gamma Q_t + 1$ with a constant $\gamma \in [0, 1)$ and $E_0^r = E_\rho(\mathbf{x}^0)$, $Q_0 = 1$; see [54]. Such $\tau_t$ can be found by adopting the simple backtracking line search technique starting from an initial guess of the stepsize $\tau_t^{(0)}$. Owing to the excellent performance of the BB method in Riemannian optimization [50, 32, 23, 25, 31, 24], we choose the initial guess $\tau_t^{(0)}$ for $t \geq 1$ as a new Riemannian BB stepsize with safeguards:

$$\tau_t^{(0)} = \min\{\max\{\tau_t^{\mathrm{BB}}, \tau_{\min}\}, \tau_{\max}\}, \quad (26)$$

where $\tau_{\max} > \tau_{\min} > 0$ are preselected stepsize safeguards and $\tau_1^{\mathrm{BB}} = \tau_1^{\mathrm{BB2}}$ and for $t \geq 2$, we set $\tau_t^{\mathrm{BB}} = \min\{\tau_{t-1}^{\mathrm{BB2}}, \tau_t^{\mathrm{BB2}}, \max\{\tau_t^{\mathrm{new}}, 0\}\}$ if $\tau_t^{\mathrm{BB2}} < \varkappa_t \tau_t^{\mathrm{BB1}}$ and set $\tau_t^{\mathrm{BB}} = \tau_t^{\mathrm{BB1}}$ otherwise. Here, $\tau_t^{\mathrm{BB1}} = \|U^t - U^{t-1}\|_{\mathsf{F}}^2 / |\langle U^t - U^{t-1}, \xi^t - \xi^{t-1}\rangle|$, $\tau_t^{\mathrm{BB2}} = |\langle U^t - U^{t-1}, \xi^t - \xi^{t-1}\rangle| / \|\xi^t - \xi^{t-1}\|_{\mathsf{F}}^2$, and $\tau_t^{\mathrm{new}}$ is chosen according to [30, Eq. (2.15)]. In our numerical tests, we set the initial $\varkappa_t$ to be 0.05 and update $\varkappa_{t+1} = \varkappa_t/1.02$ if $\tau_t^{\mathrm{BB2}}/\tau_t^{\mathrm{BB1}} < \varkappa_t$ and update $\varkappa_{t+1} = 1.02\varkappa_t$ otherwise.

We are ready to summarize the complete iRBBS in Algorithm 2. The overall complexity of Algorithm 2 to find an $(\epsilon_1, \epsilon_2)$-stationary point of problem (1) is in the same order as that of R(A)BCD.

**Theorem 3.2.** *By choosing $\epsilon_1 = \epsilon_1'/2$, $\epsilon_2 = \min\{\epsilon_1'/(4\|C\|_\infty), \epsilon_2'/(4\eta_k \Psi + 6\|C\|_\infty)\}$ with $\Psi = \|\log \pi^k\|_{\mathrm{var}} + \max\{\|\log r\|_{\mathrm{var}}, \|\log c\|_{\mathrm{var}}\}$ and $\eta_k = \epsilon_2'/(4 \log n + 2\|\log \pi^k\|_{\mathrm{var}})$, Algorithm 2 can return an $(\epsilon_1', \epsilon_2')$-stationary point of problem (1) in $\mathcal{O}(T_{\epsilon_1', \epsilon_2'})$ iterations with*

$$T_{\epsilon_1', \epsilon_2'} = \max\left\{(\epsilon_1')^{-2}, (\epsilon_2')^{-2}\right\}(\epsilon_2')^{-1}.$$

*If $\theta_t \geq 2R^t/(\eta_k(\ell_{\max} - 2 + \sqrt{2}))$ with $\ell_{\max} \geq 1$ and $R^t = \|C(U^t)\|_{\mathrm{var}} + \eta_k \Psi$, the total number of Sinkhorn iterations is $\mathcal{O}(\ell_{\max} T_{\epsilon_1', \epsilon_2'})$ and the total arithmetic operation complexity is $\mathcal{O}((n^2(k + \ell_{\max}) + ndk + dk^2)T_{\epsilon_1', \epsilon_2'})$.*

---

**Algorithm 2:** A practical iRBBS for solving problem (11).

---

1  **Input:** Choose $\tau_{\max} > \tau_{\min} > 0$, $\tau_0^{(0)} > 0$, $\epsilon_1, \epsilon_2 \geq 0$, $\sigma, \delta_1 \in (0,1)$, $\rho \in [0, \eta_k/2)$, $\gamma \in [0,1)$, and $(\alpha^{-1}, \beta^{-1}, U^0) \in \mathcal{M}$. Set $\alpha^{(0)} = \alpha^{-1}, \beta^{(0)} = \beta^{-1}$ and perform the Sinkhorn iteration (21) at $U^0$ until (23) holds with $\theta_0 = 1$ for some $\ell$. Set $\alpha^0 = \alpha^{(\ell+1)}, \beta^0 = \beta^{(\ell+1)}$.

2  **for** $t = 0, 1, \ldots$ **do**

3     Compute $\xi^t = \mathrm{grad}_U \, \mathcal{L}_{\eta_k}(\mathbf{x}^t, \pi^k)$;

4     **if** $\|\xi^t\|_{\mathsf{F}} \leq \epsilon_1$ *and* $\mathsf{e}_{\eta_k}^2(\mathbf{x}^t, \pi^k) \leq \epsilon_2$ **then return** $\mathbf{x}^t$;

5     **for** $s = 0, 1, \ldots$ **do**

6         Set $U^{t+1} = \mathrm{Retr}_{U^t}\left(-\tau_t \xi^t\right)$ with $\tau_t = \tau_t^{(0)} \sigma^s$ and update $\theta_{t+1}$ (e.g., via (27) further ahead);

7         Set $\alpha^{(0)} = \alpha^t$ and $\beta^{(0)} = \beta^t$ and perform the Sinkhorn iteration (21) at $U^{t+1}$ until (23) holds for some $\ell$; set $\alpha^{t+1} = \alpha^{(\ell+1)}$ and $\beta^{t+1} = \beta^{(\ell+1)}$.

8         **if** (25) *holds* **then break**;

---

*Remark* 3.3. The basic idea of proposing Algorithm 2 is sharply different from that of R(A)BCD developed in [28]. Ours is based on the inexact RGD viewpoint, while the latter is based on the BCD approach. Such an inexact RGD viewpoint enables us to choose the stepsize adaptively via leveraging the efficient BB stepsize. Actually, tuning the best stepsize for the $U$-update in R(A)BCD is nontrivial. It is remarked in [28, Remark 4.1] that *"the adaptive algorithms RABCD and RAGAS are also sensitive to the step size, though they are usually faster than their non-adaptive versions RBCD and RGAS."* Numerical results in Section 4.1 show the higher efficiency of our iRBBS over R(A)BCD.

*Remark* 3.4. Although the inexact gradient type methods have been well explored in the Euclidean case [11, 26, 46, 17, 39, 7, 52], to our best knowledge, there are little results for the Riemannian case and on how to choose the stepsizes adaptively for general nonlinear objective functions. One exception is R(A)GAS proposed in [34], which can be understood as the inexact RGD method. However, it needs to compute the inexact Riemannian gradient with relatively high accuracy and essentially uses the constant-like stepsizes. In contrast, our iRBBS allows to compute the inexact Riemannian gradient with low accuracy and choose the stepsize adaptively.

*Remark* 3.5. It might be better to use possible multiple Sinkhorn iterations rather than only one iteration as done in R(A)BCD in updating $\alpha$ and $\beta$ from the computational point of view. The cost of updating $\alpha^{(l+1)}$ and $\beta^{(\ell+1)}$ via one Sinkhorn iteration (21) is $\mathcal{O}(n^2)$. In contrast, the cost of updating $U^{t+1}$ via performing a RGD step (19a) is $\mathcal{O}(ndk + n^2 k + dk^2)$, wherein the main cost is to compute $V_{\phi_k(\mathbf{x}^t, \pi^k)} U^t$, which can be done by observing $V_\pi U = X \, \mathrm{Diag}(\pi \mathbf{1}) X^\mathsf{T} U + Y \, \mathrm{Diag}(\pi^\mathsf{T} \mathbf{1}) Y^\mathsf{T} U - X \pi Y^\mathsf{T} U - Y \pi^\mathsf{T} X^\mathsf{T} U$. Considering that the cost of updating $\alpha$ and $\beta$ is much less than that of updating $U$, it is reasonable to update $\alpha$ and $\beta$ multiple times and update $U$ only once.

## 4  Experimental Results

In this section, we conduct numerical experiments on six Shakespeare operas to evaluate the performance of our proposed approaches; see Dataset C.1 for a more detailed description of the dataset. All methods are implemented in MATLAB. More numerical results can be found in Appendix D. The codes are available from `https://github.com/bjiangopt/ReALM`.

### 4.1  Comparison with R(A)BCD on Solving Subproblem (11)

Subproblem (11) with $\pi^k = \mathbf{1}\mathbf{1}^\mathsf{T}$ and a relatively small $\eta_k = \eta$ is used to compute the PRW distance in [28]. We choose $\eta = 0.1$ as done in [28]. Since [28] has shown the superiority of R(A)BCD over R(A)GAS proposed in [34], we mainly compare our proposed iRBBS, namely, Algorithm 2, with R(A)BCD. For iRBBS and R(A)BCD, we use the same stopping conditions with $\epsilon_1 = 2\|C\|_\infty \epsilon_2$ (motivated by (13a)) and $\epsilon_2 = 10^{-6} \max\{\|r\|_\infty, \|c\|_\infty\}$. To make the residual error more comparable, we choose

$$\theta_0 = 1, \quad \theta_{t+1} = \max\left\{\theta \cdot \frac{\mathsf{e}_{\eta_k}^1(\mathbf{x}^t, \pi^k)}{\epsilon_1}, 1\right\} \epsilon_2, \quad t \geq 0, \tag{27}$$

Table 1: Comparison results of R(A)BCD and iRBBS for Dataset C.1. Here, "a", "b" stand for RBCD and RABCD, respectively; "c", "d", and "e" stand for iRBBS-inf, iRBBS-0.1, and iRBBS-0, respectively. For RBCD, RABCD, and iRBBS-inf, nGrad is equal to nSk.

| data | $\widehat{\mathcal{P}}_k^2$ | | | | | nGrad/nSk | | | | | time | | | | |
|---|---|---|---|---|---|---|---|---|---|---|---|---|---|---|---|
| | a | b | c | d | e | a | b | c | d | e | a | b | c | d | e |
| H5/H | **0.049074** | 0.049074 | 0.049074 | 0.049074 | 0.049074 | 871 | 604 | 320 | 64/1296 | 61/3218 | 9.6 | 6.7 | 3.4 | **1.2** | 1.8 |
| H5/JC | 0.059576 | 0.059576 | **0.059576** | 0.059576 | 0.059576 | 904 | 497 | 193 | 63/1571 | 60/3047 | 5.3 | 2.9 | 1.2 | **0.6** | 0.7 |
| H5/MV | 0.062515 | 0.062515 | 0.062515 | **0.062515** | 0.062515 | 800 | 444 | 118 | 63/1373 | 61/2812 | 5.4 | 2.9 | 0.8 | **0.6** | 0.8 |
| H5/O | 0.049973 | 0.049973 | 0.049973 | 0.049973 | **0.049973** | 612 | 397 | 129 | 43/1084 | 45/2410 | 4.9 | 3.2 | 1.0 | **0.5** | 0.7 |
| H5/RJ | 0.180227 | 0.180226 | 0.180227 | **0.180227** | 0.180227 | 474 | 440 | 319 | 73/1670 | 69/3041 | 3.8 | 3.5 | 2.4 | **0.8** | 1.0 |
| H/JC | 0.050081 | 0.050081 | 0.050081 | **0.057156** | 0.057156 | 637 | 523 | 427 | 41/1049 | 42/2020 | 4.4 | 3.6 | 2.9 | **0.4** | 0.6 |
| H/MV | **0.038416** | 0.038416 | 0.038416 | 0.038415 | 0.038416 | 2794 | 1661 | 440 | 117/2045 | 125/5129 | 22.9 | 13.6 | 3.4 | **1.2** | 1.6 |
| H/O | 0.014042 | **0.014042** | 0.014042 | 0.014042 | 0.014042 | 796 | 503 | 212 | 49/1471 | 53/3093 | 7.7 | 4.7 | 2.0 | **0.8** | 1.2 |
| H/RJ | 0.189475 | 0.189475 | 0.189475 | **0.189475** | 0.189475 | 360 | 441 | 265 | 46/933 | 47/1763 | 3.4 | 4.2 | 2.3 | **0.6** | 0.8 |
| JC/MV | 0.013257 | 0.013257 | 0.013256 | **0.013257** | 0.013257 | 1816 | 975 | 291 | 66/1781 | 65/3742 | 7.2 | 3.8 | 0.9 | **0.4** | 0.6 |
| JC/O | 0.009835 | 0.009835 | **0.009835** | 0.009835 | 0.009835 | 1160 | 763 | 101 | 52/1402 | 45/2454 | 6.5 | 3.5 | 0.5 | **0.4** | 0.5 |
| JC/RJ | 0.110336 | 0.110336 | 0.110336 | 0.110336 | **0.110336** | 340 | 332 | 235 | 44/1016 | 53/2536 | 1.5 | 1.5 | 1.1 | **0.3** | 0.5 |
| MV/O | 0.010090 | 0.010090 | **0.010090** | 0.010090 | 0.010090 | 2222 | 1396 | 154 | 90/2448 | 88/5186 | 13.2 | 8.2 | 0.9 | **0.8** | 1.1 |
| MV/RJ | 0.168293 | **0.168293** | 0.168293 | 0.168293 | 0.168293 | 842 | 763 | 632 | 85/1688 | 95/4805 | 5.0 | 4.5 | 3.8 | **0.7** | 1.1 |
| O/RJ | 0.125614 | 0.125614 | **0.125614** | 0.124085 | 0.124085 | 376 | 372 | 210 | 55/1070 | 62/2749 | 2.5 | 2.4 | 1.4 | **0.5** | 0.8 |
| AVG | 0.075387 | 0.075387 | 0.075387 | **0.075757** | 0.075757 | 1000 | 674 | 265 | 63/1460 | 65/3200 | 6.89 | 4.62 | 1.86 | **0.66** | 0.90 |

where the parameter $\theta$ is a preselected constant in the proposed iRBBS algorithm. We consider several different values of $\theta$ and denote the corresponding version of iRBBS as iRBBS-$\theta$. When $\theta = 0$, we have $\theta_{t+1} \equiv \epsilon_2$ for all $t \geq 0$, which means that we calculate $\operatorname{grad} q(U^t)$ almost exactly; when $\theta = +\infty$, the Sinkhorn iteration always stops in one iteration. For numerical tests in this subsection, we can always perform an equivalent formulation of Sinkhorn iteration (21) as [15, 6]

$$u^{(\ell+1)} = r \oslash (Av^{(\ell)}), \quad v^{(\ell+1)} = c \oslash (A^\mathsf{T} u^{(\ell+1)}), \tag{28}$$

where $\oslash$ denotes elementwise division, $u^{(\ell)} = \exp(-\alpha^{(\ell)}/\eta_k)$, $v^{(\ell)} = \exp(-\beta^{(\ell)}/\eta_k)$, and $A \in \mathbb{R}^{n \times n}$ with $A_{ij} = \pi_{ij}^k \exp(-\langle M_{ij}, UU^\mathsf{T}\rangle/\eta_k)$. More settings and details on the implementation can be found in Appendix C.

For each instance of Dataset C.1, we randomly generate 20 initial points and report the average performance in Table 1. The term "nGrad" means the total number of calculating $\operatorname{grad}_U \mathcal{L}_{\eta_k}(\mathbf{x}^t, \pi^k)$, "nSk" means the total number of Sinkhorn iterations, and "time" represents the running time in seconds evaluated by "tic-toc" commands. Moreover, since we aim to compute the PRW distance, with $U^t$ returned by some method in hand, we invoke Gurobi 9.5/Mosek 9.3 to compute a more accurate PRW distance, denoted as $\widehat{\mathcal{P}}_k^2 = \min_{\pi \in \Pi(r,c)} \langle \pi, C(U^t)\rangle$. Note that a larger $\widehat{\mathcal{P}}_k^2$ means a higher solution quality of the corresponding $U^t$. In the last "AVG" line, we summarize the averaged performance.

From Table 1, we have the following observations: (i) as for the solution quality, iRBBS-0.1 performs the best among all methods; (ii) among iRBBS methods, iRBBS-0.1 is about 2.8x faster than iRBBS-inf and about 1.4x faster than iRBBS-0; (iii) compared with R(A)BCD, iRBBS-0.1 can always take significantly less nGrad and may take a bit more nSk. This makes it about 7x faster than RABCD and about 10.4x faster than RBCD. The results show that iRBBS generally performs much better than R(A)BCD. More importantly, our methods adopt the adaptive stepsize without needing to tune the best stepsize as done in R(A)BCD.

## 4.2 Comparison on Computing the PRW Distance (1)

In this subsection, we present numerical results to illustrate the effectiveness and efficiency of our proposed REALM, namely, Algorithm 1. We choose $\epsilon_1 = 2\|C\|_\infty \epsilon_2$, $\epsilon_2 = 10^{-6} \max\{\|r\|_\infty, \|c\|_\infty\}$, $\epsilon_c = 10^{-3}$, and $\eta_1 = 20$. To avoid possible numerical instability, we restrict the maximum number of updating $\pi^k$ as 8. To prevent too small $\eta_k$, in our implementation, we update $\eta_{k+1} = \min\{\gamma_\eta \eta_k, \eta_{\min}\}$ other than using (16), where $\eta_{\min}$ is a preset positive number. We set $\gamma_\eta = \gamma_\epsilon = 0.25$. Moreover, we denote by REALM-$(\eta_{\min}, \gamma_W)$ the REALM with particular parameters $\eta_{\min}$ and $\gamma_W$. Note that choosing $\gamma_W = 0$ means that we adopt a continuation technique to solve (11) with $\pi^k \equiv \pi^1$ and $\eta_k = \eta_{\min}$, which is generally better than directly solving a single (11) with

Table 2: Average results of REALM for Dataset C.1, "a" and "b" stand for REALM-$(0.0035, 0)$ and REALM-$(0.007, 0.9)$, respectively.

| | $\widehat{\mathcal{P}}_k^2$ | | nGrad | | $\text{nSk}_{\exp}/\text{nSk}_{\log}$ | | time | | iter | |
|---|---|---|---|---|---|---|---|---|---|---|
| data | a | b | a | b | a | b | a | b | a | b |
| H5/JC | 0.09270 | **0.10985** | 1081 | 675 | 23809/8267 | 9786/0 | 84.1 | 8.6 | 0.0/8.0 | 8.0/15.0 |
| H/MV | 0.06378 | **0.06424** | 1116 | 727 | 3864/13382 | 30997/0 | 175.7 | 14.9 | 0.0/8.0 | 8.0/15.0 |
| H/RJ | 0.21706 | **0.22607** | 547 | 788 | 3091/4062 | 11747/0 | 63.3 | 15.6 | 0.0/8.0 | 8.0/15.0 |
| JC/MV | **0.06270** | 0.06235 | 1324 | 711 | 12875/9727 | 17073/0 | 55.3 | 5.9 | 0.0/8.0 | 8.0/15.0 |
| JC/O | 0.04221 | **0.04277** | 1650 | 1100 | 4997/26612 | 35031/4500 | 204.7 | 46.7 | 0.0/8.0 | 8.0/15.0 |
| MV/O | **0.04181** | 0.03661 | 890 | 822 | 6101/5289 | 31651/0 | 57.0 | 13.3 | 0.0/8.0 | 8.0/15.0 |
| AVG | 0.11338 | **0.11461** | 863 | 787 | 7824/7364 | 20261/300 | 78.0 | 15.0 | 0.0/8.0 | 8.0/15.0 |

$\eta_k = \eta_{\min}$. Subproblem (11) in REALM is solved by iRBBS. If $\max\{\|r\|_\infty, \|c\|_\infty\} \geq 500\eta_k$ or $\|C(U^t) - \eta_k \log \pi^k\|_{\text{var}} \geq 900\eta_k$, we set $\theta = 10$ and perform the Sinkhorn iteration (21); otherwise, we set $\theta = 0.1$ and perform an equivalent formulation of Sinkhorn iteration (28).

The results over 20 runs on Dataset C.1 are reported in Table 2. In this table, the terms "$\text{nSk}_{\log}$" and "$\text{nSk}_{\exp}$" mean the total numbers of Sinkhorn iterations (21) and (28), respectively, the pair "$k_1/k$" in the column "iter" means that the corresponding algorithm stops at the $k$-iteration and updates the multiplier matrix $k_1$ times. To save space, we only report instances where one method can return the value "$\widehat{\mathcal{P}}_k^2$" larger than 1.005 times of the smaller one of the two $\widehat{\mathcal{P}}_k^2$ values returned by the two methods. The better "$\widehat{\mathcal{P}}_k^2$" is marked in **bold**. Besides, the average performance over all 15 instances is also kept in the "AVG" line.

From Table 2, we can observe that REALM-$(0.007, 0.9)$ can not only return better solutions than REALM-$(0.0035, 0)$ but also is about 5.2x faster. On average, REALM-$(0.007, 0.9)$ updates the multiplier matrix 8 times in 15 total iterations, which shows that updating the multiplier matrix does help. The reasons why REALM with updating the multiplier matrix outperforms REALM without updating the multiplier matrix in terms of solution quality and speed are as follows. First, updating the multiplier matrix in REALM can keep the solution quality even using a larger $\eta_k$. Second, solving the subproblem with a larger $\eta_k$ is always easier, which enables that REALM-$(0.07, 0.9)$ computes less $\text{grad}_U \mathcal{L}_{\eta_k}(\mathbf{x}^t, \pi^k)$ and performs less Sinkhorn iterations (21) which involves computing the log-sum-exp function $\log \sum_i \exp(x_i/\eta_k) = x_{\max}/\eta_k + \log \sum_i \exp((x_i - x_{\max})/\eta_k)$ for small $\eta_k$.

## 5    Concluding Remarks

In this paper, we considered the computation of the PRW distance. By reformulating this problem as an optimization problem over the Cartesian product of the Stiefel manifold and the Euclidean space with additional nonlinear inequality constraints, we proposed a method called REALM. The convergence of REALM was also established. To solve the subproblem in REALM efficiently, we developed a practical iRBBS method with convergence and iteration complexity guarantees, wherein the Riemannian BB stepsize (based on the inexact Riemannian gradient information) and Sinkhorn iterations are employed. The complexity of iRBBS to attain an $\epsilon$-stationary point of the original PRW distance problem matches the best known iteration complexity result. Numerical results showed that, compared with the state-of-the-art methods, our proposed REALM and iRBBS methods have advantages in solution quality and speed.

Moreover, our proposed REALM and iRBBS can also be extended to solve some important mini-max problems over the Riemannian manifolds arising from machine learning, such as the fair PCA problem [45] and the projection robust Wasserstein barycenters [27], etc.

Lastly, a limitation of our work is that we did not establish the positive lower bound of $\eta_k$ in REALM, despite the fact that REALM performs well in practice and can avoid too small $\eta_k$ in many cases. We shall investigate the conditions under which it is possible to establish a lower bound of $\eta_k$ in REALM. This can be achievable by extending the analysis and conditions in [20] to the Riemannian case.

**Acknowledgments** The work of Bo Jiang was supported in part by the National Natural Science Foundation of China (NSFC) under Grant 11971239, Grant 12371314, and in part by the Natural Science Foundation of the Higher Education Institutions of Jiangsu Province under Grant 21KJA110002. The work of Ya-Feng Liu was supported in part by NSFC under Grant 11991020, Grant 11991021, and Grant 12288201.

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

# A  More Details on Our Proposed REALM in Section 2

In this section, we provide more materials on REALM in Section 2, including the first-order necessary conditions of our reformulation (6), relations between different $(\epsilon_1, \epsilon_2)$-stationary points of the PRW distance problem (1), the connections of the approximate stationary points of problems (11) and (1), and the convergence analysis of Algorithm 1.

## A.1  First-Order Necessary Conditions of Problem (6)

**Lemma A.1** (First-order necessary conditions of problem (6))**.** *Given $\bar{\mathbf{x}} \in \mathcal{M}$ and $\bar{y} \in \mathbb{R}$, suppose that $(\bar{\mathbf{x}}, \bar{y})$ is a local minimizer of problem* (6)*, then $(\bar{\mathbf{x}}, -\varphi(\bar{\mathbf{x}})_{\min})$ is a stationary point of problem* (6)*, namely, there exists $\bar{\pi} \in \Pi(r, c)$ such that*

$$\mathsf{Proj}_{\mathrm{T}_{\bar{U}}\mathcal{U}} \left( -2V_{\bar{\pi}}\bar{U} \right) = 0, \quad \langle \bar{\pi}, Z(\bar{\mathbf{x}}) \rangle = 0, \tag{29}$$

*where $Z(\bar{\mathbf{x}}) \in \mathbb{R}^{n \times n}$ is defined as*

$$Z(\bar{\mathbf{x}})_{ij} = \varphi(\bar{\mathbf{x}})_{ij} - \varphi(\bar{\mathbf{x}})_{\min}. \tag{30}$$

*Proof.* Since $(\bar{\mathbf{x}}, \bar{y})$ is a local minimizer of problem (6), there must hold $\bar{y} = -\varphi(\bar{\mathbf{x}})_{\min} = \max_{ij}\{-\varphi(\bar{\mathbf{x}})_{ij}\}$. Moreover, such $\bar{\mathbf{x}}$ is also a local minimizer of problem $\min_{\mathbf{x} \in \mathcal{M}} h(\mathbf{x})$ with $h(\mathbf{x}) = r^{\mathsf{T}}\alpha + c^{\mathsf{T}}\beta - \varphi(\mathbf{x})_{\min}$. For fixed $\alpha$ and $\beta$, it is easy to see that $-\varphi(\mathbf{x})_{ij} + \|C\|_{\infty}\|U\|_{\mathsf{F}}^2$ is convex with respect to $U$ since $\|M_{ij}\|_2 = C_{ij} \leq \|C\|_{\infty}$. We thus know that the function

$$-\varphi(\bar{\mathbf{x}})_{\min} + \|C\|_{\infty}\|U\|_{\mathsf{F}}^2 = \max_{ij} \left\{ -\varphi(\mathbf{x})_{ij} + \|C\|_{\infty}\|U\|_{\mathsf{F}}^2 \right\}$$

is convex with respect to $U$, which means that the function $-\varphi(\bar{\mathbf{x}})_{\min}$ and thus $h(\mathbf{x})$ is $\|C\|_{\infty}$-weakly convex with respect to $U$ [48, Proposition 4.3]. Let $\mathcal{A}(\bar{\mathbf{x}}) = \{(i, j) \in [n] \times [n] \mid \varphi(\bar{\mathbf{x}})_{ij} = \varphi(\bar{\mathbf{x}})_{\min}\}$. By [48, Proposition 4.6], we have

$$\partial h(\bar{\mathbf{x}}) = (r, c, 0) + \mathrm{conv}\left\{ (-e_i, -e_j, -2M_{ij}\bar{U}) \mid (i, j) \in \mathcal{A}(\bar{\mathbf{x}}) \right\},$$

where $e_i$ is the $i$-th standard unit vector in $\mathbb{R}^n$. Moreover, by [53, Theorem 4.1], [53, Theorem 5.1], and $\mathrm{T}_{\bar{\mathbf{x}}}\mathcal{M} = \mathbb{R}^n \times \mathbb{R}^n \times \mathrm{T}_{\bar{\mathbf{x}}}\mathcal{U}$, there must hold that $0 \in \mathsf{Proj}_{\mathrm{T}_{\bar{\mathbf{x}}}\mathcal{M}}\partial h(\bar{\mathbf{x}})$. Putting all the above things together shows that there exists $\bar{\pi} \in \mathbb{R}_+^{n \times n}$ with $\bar{\pi}_{ij} = 0$ for all $(i, j) \notin \mathcal{A}(\bar{\mathbf{x}})$ and $\sum_{(i,j) \in \mathcal{A}(\bar{\mathbf{x}})} \bar{\pi}_{i,j} = 1$ such that $\bar{\pi} \in \Pi(r, c)$ and (29) hold. The proof is completed. $\qquad\square$

## A.2  Explanations on Remark 2.2

From [28, Section B], we know that the $(\epsilon_1, \epsilon_2)$-stationary point in [28, Definition 3.1] is stronger than that in [34, Definition 2.7]. To show that our Definition 2.1 of the $(\epsilon_1, \epsilon_2)$-stationary point is stronger than both definitions, we only need to verify that $\langle \tilde{\pi}, Z(\tilde{\mathbf{x}}) \rangle \leq \epsilon_2$ can imply

$$\langle \tilde{\pi}, C(\tilde{U}) \rangle - \min_{\pi \in \Pi(r,c)} \langle \pi, C(\tilde{U}) \rangle \leq \epsilon_2,$$

which is clear since $\langle \tilde{\pi}, C(\tilde{U}) \rangle - \min_{\pi \in \Pi(r,c)} \langle \pi, C(\tilde{U}) \rangle \leq \langle \tilde{\pi}, Z(\tilde{\mathbf{x}}) \rangle$. This inequality comes from the fact that problem (5) is a dual formulation of $\min_{\pi \in \Pi(r,c)} \langle \pi, C(U) \rangle$ for fixed $U$ and the strong duality theorem.

## A.3  Proof of Theorem 2.4

*Proof.* We first show that (13a) is true. First, by [12, Lemma 3.2], we have $\hat{\pi} \in \Pi(r, c)$ and

$$\|\hat{\pi} - \phi_{\eta_k}(\tilde{\mathbf{x}}, \pi^k)\|_1 \leq \|\phi_{\eta_k}(\tilde{\mathbf{x}}, \pi^k)\mathbf{1} - r\|_1 + \|\phi_{\eta_k}(\tilde{\mathbf{x}}, \pi^k)^{\mathsf{T}}\mathbf{1} - c\|_1 = \mathsf{e}_{\eta_k}^2(\tilde{\mathbf{x}}, \pi^k) \leq \epsilon_2. \tag{31}$$

Second, by the triangular inequality and the non-expansive property of the projection operator, we have
$$\|\mathrm{Proj}_{\mathrm{T}_U\mathcal{U}}(A_1)\|_\mathsf{F} \le \|\mathrm{Proj}_{\mathrm{T}_U\mathcal{U}}(A_1) - \mathrm{Proj}_{\mathrm{T}_U\mathcal{U}}(A_2)\|_\mathsf{F} + \|\mathrm{Proj}_{\mathrm{T}_U\mathcal{U}}(A_1)\|_\mathsf{F}$$
$$\le \|A_1 - A_2\|_\mathsf{F} + \|\mathrm{Proj}_{\mathrm{T}_U\mathcal{U}}(A_1)\|_\mathsf{F},$$
where $A_1, A_2 \in \mathbb{R}^{d\times k}$. Hence, we have
$$\|\mathrm{Proj}_{\mathrm{T}_U\mathcal{U}}(-2V_{\hat\pi}\tilde{U})\|_\mathsf{F} \le 2\|(V_{\hat\pi} - V_{\phi_{\eta_k}(\tilde{\mathbf{x}},\pi^k)})\tilde{U}\|_\mathsf{F} + \epsilon_1 \le 2\|V_{\hat\pi} - V_{\phi_{\eta_k}(\tilde{\mathbf{x}},\pi^k)}\|_\mathsf{F} + \epsilon_1, \qquad (32)$$
where the first inequality uses $\|\mathrm{Proj}_{\mathrm{T}_{\tilde{U}}\mathcal{U}}(-2V_{\phi_{\eta_k}(\tilde{\mathbf{x}},\pi^k)}\tilde{U})\|_\mathsf{F} \le \epsilon_1$ and the second inequality is due to the fact that $\|A\tilde{U}\|_\mathsf{F} \le \|A\|_\mathsf{F}$ for any matrix $A \in \mathbb{R}^{d\times d}$. Moreover, observe that
$$\|V_{\hat\pi} - V_{\phi_{\eta_k}(\tilde{\mathbf{x}},\pi^k)}\|_\mathsf{F} = \Big\| \sum_{ij} (\hat\pi_{ij} - \phi_{\eta_k}(\tilde{\mathbf{x}},\pi^k)_{ij})M_{ij} \Big\|_\mathsf{F} \le \|C\|_\infty \|\hat\pi - \phi_{\eta_k}(\tilde{\mathbf{x}},\pi^k)\|_1.$$

Combining the above assertions with (31) and (32) yields (13a).

Next, we show that (13b) is also true. By the Cauchy-Schwarz inequality and (31), we have
$$\langle \hat\pi, Z(\tilde{\mathbf{x}}) \rangle = \langle \hat\pi - \phi_{\eta_k}(\tilde{\mathbf{x}},\pi^k), Z(\tilde{\mathbf{x}}) \rangle + \langle \phi_{\eta_k}(\tilde{\mathbf{x}},\pi^k), Z(\tilde{\mathbf{x}}) \rangle$$
$$\le \|Z(\tilde{\mathbf{x}})\|_\infty \epsilon_2 + \langle \phi_{\eta_k}(\tilde{\mathbf{x}},\pi^k), Z(\tilde{\mathbf{x}}) \rangle$$
$$\le (\|\tilde\alpha\|_{\mathrm{var}} + \|\tilde\beta\|_{\mathrm{var}} + \|C\|_\infty)\epsilon_2 + \langle \phi_{\eta_k}(\tilde{\mathbf{x}},\pi^k), Z(\tilde{\mathbf{x}}) \rangle, \qquad (33)$$
where the second inequality uses (30), the definition of $\varphi(\mathbf{x})$ in (5), and
$$\langle M_{ij}, UU^\mathsf{T} \rangle = [C(U)]_{ij} \le C_{ij} \le \|C\|_\infty \quad \forall\, U \in \mathcal{U}. \qquad (34)$$
The remaining is to bound $\langle \phi_{\eta_k}(\tilde{\mathbf{x}},\pi^k), Z(\tilde{\mathbf{x}}) \rangle$. By $\|\zeta_{\eta_k}(\tilde{\mathbf{x}},\pi^k)\|_1 = 1$, (7), and (12), we have $\varphi(\tilde{\mathbf{x}})_{ij} = \eta_k(\log \pi_{ij}^k - \log \phi_{\eta_k}(\tilde{\mathbf{x}},\pi^k)_{ij})$ and $\varphi(\tilde{\mathbf{x}})_{\min} \ge \eta_k(\log \pi^k)_{\min}$. Again with (30), we have
$$\langle \phi_{\eta_k}(\tilde{\mathbf{x}},\pi^k), Z(\tilde{\mathbf{x}}) \rangle = \langle \phi_{\eta_k}(\tilde{\mathbf{x}},\pi^k), \varphi(\tilde{\mathbf{x}}) \rangle - \varphi(\tilde{\mathbf{x}})_{\min}$$
$$\le \eta_k \left( \langle \log \pi^k, \phi_{\eta_k}(\tilde{\mathbf{x}},\pi^k) \rangle + H(\phi_{\eta_k}(\tilde{\mathbf{x}},\pi^k)) - (\log \pi^k)_{\min} \right),$$
which, together with $H(\phi_{\eta_k}(\tilde{\mathbf{x}},\pi^k)) \le 2\log n$ and $\langle \log \pi^k, \phi_{\eta_k}(\tilde{\mathbf{x}},\pi^k) \rangle \le (\log \pi^k)_{\max}$, further implies $\langle \phi_{\eta_k}(\tilde{\mathbf{x}},\pi^k), Z(\tilde{\mathbf{x}}) \rangle \le (2\log n + \|\log \pi^k\|_{\mathrm{var}})\eta_k$. This, together with (33), yields (13b). The proof is completed. $\qquad\square$

### A.4 Proof of Theorem 2.7

We first use the requirements in (14) to show that $r^\mathsf{T}\alpha^k \le r^\mathsf{T}\alpha^0$ for all $k \ge 0$.

With the first two requirements in (14), we have $\mathcal{L}_{\eta_k}(\mathbf{x}^k, \pi^k) = 2r^\mathsf{T}\alpha^k$. By the fact $\|\zeta_{\eta_1}(\mathbf{x}^0, \pi^1)\|_1 = 1$ with $\pi^1 = \mathbf{1}\mathbf{1}^\mathsf{T}$, we know from (7) that $\varphi(\mathbf{x}^0)_{ij} \ge 0$. Due to the update rules of $\pi^k$ and $\eta_k$, we know that $0 < \pi_{ij}^k \le 1$ and $0 < \eta_k \le \eta_1$ for all $k \ge 1$. Therefore, we have from (7) that
$$[\zeta_{\eta_k}(\mathbf{x}^0, \pi^k)]_{ij} = \pi_{ij}^k \exp\left( -\frac{\varphi(\mathbf{x}^0)_{ij}}{\eta_k} \right) \le \exp\left( -\frac{\varphi(\mathbf{x}^0)_{ij}}{\eta_1} \right) = [\zeta_{\eta_1}(\mathbf{x}^0, \pi^1)]_{ij},$$
which further implies $\|\zeta_{\eta_k}(\mathbf{x}^0, \pi^k)\|_1 \le \|\zeta_{\eta_1}(\mathbf{x}^0, \pi^1)\|_1 = 1$. With $r^\mathsf{T}\alpha^0 = c^\mathsf{T}\beta^0$, we know $\mathcal{L}_{\eta_k}(\mathbf{x}^0, \pi^k) \le 2r^\mathsf{T}\alpha^0$. By the third requirement in (14), we further have
$$r^\mathsf{T}\alpha^k \le r^\mathsf{T}\alpha^0, \quad \forall\, k \ge 0. \qquad (35)$$
Besides, since $\mathbf{x}^k$ is an $(\epsilon_{k,1}, \epsilon_{k,2})$-stationary point of (11), we have
$$\left\|\mathrm{Proj}_{\mathrm{T}_{U^k}\mathcal{U}}\left(-2V_{\tilde\pi^{k+1}}U^k\right)\right\|_\mathsf{F} \le \epsilon_{k,1}, \quad \|r - \tilde\pi^{k+1}\mathbf{1}\|_1 + \|c - (\tilde\pi^{k+1})^\mathsf{T}\mathbf{1}\|_1 \le \epsilon_{k,2}. \qquad (36)$$

We next consider two cases, which is motivated by the proofs of Theorem 3.1 in [13].

Case (i). $\eta_k$ is bounded below by some threshold value $\underline{\eta} > 0$. Due to the update rule of $\eta_{k+1}$, we can see that (16) is invoked only finite times. Besides, due to $0 < \pi_{ij}^k \le 1$, without loss of generality,

we assume $\eta_k \equiv \eta$ for all $k \geq 2$ and $\lim_{k \to \infty} \pi_{ij}^k = \pi_{ij}^\infty$. Meanwhile, due to the update rule of $\pi^k$, we have $\pi_{ij}^k = \tilde{\pi}_{ij}^k$ for $k \geq 2$ and

$$\left|\min\{\underline{\eta}\pi_{ij}^{k+1}, \varphi(\mathbf{x}^k)_{ij}\}\right| \to 0 \quad \forall (i,j) \in [n] \times [n]. \tag{37}$$

We now show that $\{\mathbf{x}^k\}$ has at least a limit point. By (37), we must have $\varphi(\mathbf{x}^k)_{ij} \to 0$ for each $(i,j) \notin \mathcal{A}(\pi^\infty) := \{(i,j) \in [n] \times [n] \mid \pi_{ij}^\infty = 0\}$ and there must exist $K_1 > 0$ such that for all $k \geq K_1$, there holds $\varphi(\mathbf{x}^k)_{ij} \geq -1$ for each $(i,j) \in \mathcal{A}(\pi^\infty)$. Recalling the definition of $\varphi(\mathbf{x})$ in (5), then we can conclude that there exists $K_2 > 0$ such that for all $k \geq K_2$, there holds that

$$\alpha_i^k + \beta_j^k + \langle M_{ij}, U^k(U^k)^\mathsf{T}\rangle \geq -1 \quad \forall (i,j) \in [n] \times [n].$$

Multiplying both sides of the above assertions by $c_j$ and then summing the obtained inequality from $j = 1$ to $n$, we have

$$\alpha_i^k \geq -1 - \sum_j c_j \langle M_{ij}, U^k(U^k)^\mathsf{T}\rangle - c^\mathsf{T}\beta^k \geq -1 - \|C\|_\infty - r^\mathsf{T}\alpha^0, \tag{38}$$

where the second inequality is due to $r^\mathsf{T}\alpha^k = c^\mathsf{T}\beta^k$, (35), and (34). Combining (35) and (38), we further have

$$-1 - \|C\|_\infty - r^\mathsf{T}\alpha^0 \leq \alpha_i^k \leq r_i^{-1}\big((1-r_i)(1 + \|C\|_\infty + r^\mathsf{T}\alpha^0) + r^\mathsf{T}\alpha^0\big).$$

Similarly, we can establish a similar bound for each $\beta_j^k$. Recalling that $U^k$ is in a compact set, it is thus safe to say that the sequence $\{\mathbf{x}^k\}$ has at least one limit point, denoted by $\mathbf{x}^\infty = \lim_{k \in \mathcal{K}, k \to \infty} \mathbf{x}^k$. Again with $\lim_{k \to \infty} \pi_{ij}^k = \pi^\infty$, we have from (37) that $\min\{\underline{\eta}\pi_{ij}^\infty, \varphi(\mathbf{x}^\infty)_{ij}\} = 0$, which, together with $\pi_{ij}^\infty \geq 0$ and the fact

$$0 \leq \langle \pi^\infty, Z(\mathbf{x}^\infty)\rangle = \sum_{ij} \pi_{ij}^\infty \varphi(\mathbf{x}^\infty)_{ij} - \varphi(\mathbf{x}^\infty)_{\min},$$

further implies $\varphi(\mathbf{x}^\infty)_{\min} = 0$ and $\langle \pi^\infty, Z(\mathbf{x}^\infty)\rangle = 0$. This shows that $(\mathbf{x}^\infty, 0)$ is feasible. Moreover, letting $k \in \mathcal{K}$ go to infinity in (36), we further have $\pi^\infty \in \Pi(r,c)$ and $\|\mathrm{Proj}_{\mathrm{T}_{U^\infty}\mathcal{U}}(-2V_{\pi^\infty}U^\infty)\|_\mathsf{F} = 0$. From Lemma A.1, we know that $(\mathbf{x}^\infty, 0)$ is a stationary point of problem (6) and $(\mathbf{x}^\infty, \pi^\infty)$, namely, $(\mathbf{x}^\infty, \tilde{\pi}^\infty)$, is a stationary point of problem (1).

Case (ii). The sequence $\{\eta_k\}$ is not bounded below by any positive number, namely, $\lim_{k \to \infty} \eta_k = 0$. By the updating rule, we know that $\eta_k$ is updated via (16) infinitely many times. Hence, there must exist $k_1 < k_2 < \cdots$ such that $\eta_{k_\ell} \to 0$ as $\ell \to \infty$ and

$$\eta_s = \eta_{k_\ell} = \min\{\gamma_\eta \eta_{k_\ell - 1}, \varrho_{k_\ell - 1}/\|\log \pi^{k_\ell - 1}\|_\infty\} \quad \text{with} \quad k_\ell \leq s < k_{\ell+1} \quad \text{and} \quad \pi^{k_\ell} = \pi^{k_\ell - 1}.$$

By (7) and the second assertion in (14), we have $\pi_{ij}^{k_\ell} \exp(-\varphi(\mathbf{x}^{k_\ell})_{ij}/\eta_{k_\ell}) \leq 1$, which, together with $0 < \pi_{ij}^{k_\ell} \leq 1$ and (16), implies

$$\varphi(\mathbf{x}^{k_\ell})_{ij} \geq -\eta_{k_\ell}|\log \pi_{ij}^{k_\ell}| \geq -\eta_{k_\ell}\|\log \pi^{k_\ell - 1}\|_\infty \geq -\varrho_{k_\ell - 1}, \quad \forall \ell \geq 1. \tag{39}$$

Using the same arguments as in the proof in Case (i), we can show that $\{\mathbf{x}^k\}$ is bounded over $\{k_1, k_2, \ldots\}$ and thus $\{\mathbf{x}^k\}$ has at least a limit point. Without loss of generality, assume $\lim_{\ell \to \infty} \mathbf{x}^{k_\ell} = \mathbf{x}^\infty$. By (39) and $\varrho_k \to 0$ in (16), we know that $\varphi(\mathbf{x}^\infty) \geq 0$ and thus that $(\mathbf{x}^\infty, 0)$ is feasible to problem (6).

Due to the compactness of $\tilde{\pi}^k$ in (15), there must exist a subset $\mathcal{K} \subseteq \{k_1, k_2, \ldots\}$ such that $\lim_{k \in \mathcal{K}, k \to \infty} \tilde{\pi}_{ij}^k = \tilde{\pi}^\infty$. Recalling (15), (7), and the second requirement in (14), we have $\tilde{\pi}_{ij}^{k+1} = \pi_{ij}^k \exp(-\varphi(\mathbf{x}^k)_{ij}/\eta_k)$ for each $k \geq 1$. Let $\mathcal{A}(\mathbf{x}^\infty) = \{(i,j) \in [n] \times [n] \mid \varphi(\mathbf{x}^\infty)_{ij} = 0\}$. We claim $\mathcal{A}(\mathbf{x}^\infty) \neq \emptyset$. Otherwise, for every $(i,j)$ we have $\varphi(\mathbf{x}^\infty)_{ij} > 0$ and thus

$$0 \leq \lim_{k \in \mathcal{K}, k \to \infty} \tilde{\pi}_{ij}^{k+1} \leq \lim_{k \in \mathcal{K}, k \to \infty} \exp\left(-\frac{\varphi(\mathbf{x}^k)_{ij}}{\eta_k}\right) = 0, \tag{40}$$

where the equality uses $\lim_{\ell \to \infty} \mathbf{x}^{k_\ell} = \mathbf{x}^\infty$, $\mathcal{K} \subseteq \{k_1, k_2, \ldots\}$, and $\lim_{k \to \infty} \eta_k \to 0$. This makes a contradiction with $\|\tilde{\pi}^{k+1}\|_1 = 1$. Moreover, (40) also further implies that $\langle \tilde{\pi}^\infty, Z(\mathbf{x}^\infty)\rangle = 0$. Finally, with (36), we further have $\|\mathrm{Proj}_{\mathrm{T}_{U^\infty}\mathcal{U}}(-2V_{\tilde{\pi}^\infty}U^\infty)\|_\mathsf{F} = 0$ and $\tilde{\pi}^\infty \in \Pi(r,c)$. Putting the above things together, we know that that $(\mathbf{x}^\infty, 0)$ is a stationary point of (6) and $(\mathbf{x}^\infty, \tilde{\pi}^\infty)$ is a stationary point of problem (1). The proof is completed.

# B Proofs in Section 3

## B.1 Proof of Lemma 3.1

For notational simplicity, we define

$$h(\pi, U) = \langle C(U) - \eta_k \log \pi^k, \pi \rangle - \eta_k H(\pi). \tag{41}$$

It is easy to see that

$$\max_{\alpha \in \mathbb{R}^n, \beta \in \mathbb{R}^n} -\mathcal{L}_{\eta_k}(\mathbf{x}, \pi^k) \quad \text{is the dual formulation of} \quad \min_{\pi \in \Pi(r,c)} h(\pi, U). \tag{42}$$

Therefore, we can write

$$q(U) = -\min_{\pi \in \Pi(r,c)} h(\pi, U). \tag{43}$$

Since the entropy function is 1-strongly convex with respect to the $\ell_1$-norm over the probability simplex, the minimization problem in (43) has a unique solution $\pi_U^* = \operatorname{argmin}_{\pi \in \Pi(r,c)} h(\pi, U)$. Using [34, Lemma B.1], we thus have

$$\operatorname{grad} q(U) = \operatorname{Proj}_{\mathrm{T}_U \mathcal{U}}(-2V_{\pi_U^*} U). \tag{44}$$

By (5), (7), (12), and recalling $[C(U)]_{ij} = \langle M_{ij}, UU^\mathsf{T} \rangle$, we have

$$[C(U)]_{ij} + \eta_k \log \phi_{\eta_k}(\mathbf{x}_U^*, \pi^k)_{ij} - \eta_k \log \pi_{ij}^k = -[\alpha_U^*]_i - [\beta_U^*]_j - \eta_k \log(\|\zeta_{\eta_k}(\mathbf{x}_U^*, \pi^k)\|_1). \tag{45}$$

By the optimality of $\alpha_U^*$ and $\beta_U^*$, we know that $\mathrm{e}_{\eta_k}^1(\mathbf{x}_U^*, \pi^k) = \mathrm{e}_{\eta_k}^2(\mathbf{x}_U^*, \pi^k) = 0$ and thus $\phi_{\eta_k}(\mathbf{x}_U^*, \pi^k)\mathbf{1} = r$ and $\phi_{\eta_k}(\mathbf{x}_U^*, \pi^k)^\mathsf{T}\mathbf{1} = c$. With (45), (41), and the definition of $\mathcal{L}_{\eta_k}(\cdot, \cdot)$ in (11), by some calculations, we have

$$h(\phi_{\eta_k}(\mathbf{x}_U^*, \pi^k), U) = -r^\mathsf{T}\alpha_U^* - c^\mathsf{T}\beta_U^* - \eta_k \log(\|\zeta_{\eta_k}(\mathbf{x}_U^*, \pi^k)\|_1) = -\mathcal{L}_{\eta_k}(\mathbf{x}_U^*, \pi^k).$$

By the dual formulation (42), we have $-\mathcal{L}_{\eta_k}(\mathbf{x}_U^*, \pi^k) \leq h(\pi_U^*, U)$. Hence, we know that $h(\phi_{\eta_k}(\mathbf{x}_U^*, \pi^k), U) \leq h(\pi_U^*, U)$, which, together with the optimality and uniqueness of $\pi_U^*$, yields $\phi_{\eta_k}(\mathbf{x}_U^*, \pi^k) = \pi_U^*$. Finally, using (44), we can complete the proof.

## B.2 Proof of Theorem 3.2

We first establish the iteration complexity result of iRBBS, which is based on the following nice properties of Sinkhorn iteration (21). Note that the sufficient decrease property in Lemma B.2 is quite different from that in [29, Lemmas 4.10 & 4.11] since we can establish the non-increasing property of the feasibility violation of $\pi^{(l)}$ in Lemma B.1.

**Lemma B.1.** *Let $\{(\alpha^\ell, \beta^\ell)\}$ be the sequence generated by (21) and consider $\{\pi^{(\ell)}\}$ and $\{\pi^{(\ell+\frac{1}{2})}\}$ in (22). Then we have $\|\pi^{(0)}\mathbf{1} - r\|_1 + \|(\pi^{(0)})^\mathsf{T}\mathbf{1} - c\|_1 \geq \|(\pi^{(\frac{1}{2})})^\mathsf{T}\mathbf{1} - c\|_1 \geq \|\pi^{(1)}\mathbf{1} - r\|_1$ and for each $\ell \geq 1$,*

$$\|\pi^{(\ell)}\mathbf{1} - r\|_1 \geq \|(\pi^{(\ell+\frac{1}{2})})^\mathsf{T}\mathbf{1} - c\|_1 \geq \|\pi^{(\ell+1)}\mathbf{1} - r\|_1. \tag{46}$$

*Proof.* By the optimality of $\alpha^{(\ell+1)}$ in (20) and the first equation in (22), we have $\pi^{(\ell+\frac{1}{2})}\mathbf{1} - r = 0$ for $\ell \geq 0$. Therefore, we have

$$\|\pi^{(\ell+1)}\mathbf{1} - r\|_1 = \|\pi^{(\ell+\frac{1}{2})}\mathbf{1} - \pi^{(\ell+1)}\mathbf{1}\|_1 \leq \|\pi^{(\ell+\frac{1}{2})} - \pi^{(\ell+1)}\|_1$$

$$= \sum_{ij} \pi_{ij}^{(\ell+\frac{1}{2})} \left| 1 - \exp\left(-\frac{\beta_j^{(\ell+1)} - \beta_j^{(\ell)}}{\eta_k}\right) \right|$$

$$= \sum_{ij} \frac{\pi_{ij}^{(\ell+\frac{1}{2})}}{[(\pi^{(\ell+\frac{1}{2})})^\mathsf{T}\mathbf{1}]_j} \left| [(\pi^{(\ell+\frac{1}{2})})^\mathsf{T}\mathbf{1}]_j - c_j \right| = \|(\pi^{(\ell+\frac{1}{2})})^\mathsf{T}\mathbf{1} - c\|_1,$$

where the first inequality uses the Cauchy-Schwarz inequality, the second equality comes from (7), (22), and the definitions of $\zeta^{(\ell)}$ and $\zeta^{(\ell+\frac{1}{2})}$ after (21), and the third equality is due to the second equation in (21) and the first equation in (22). This proves $\|(\pi^{(\ell+\frac{1}{2})})^\mathsf{T}\mathbf{1} - c\|_1 \geq \|\pi^{(\ell+1)}\mathbf{1} - r\|_1$

for $\ell \geq 0$. On the other hand, by the optimality of $\beta^{(\ell+1)}$ in (20) and the second equation in (22), we have $(\pi^{(\ell)})^\mathsf{T}\mathbf{1} - c = 0$ for $\ell \geq 1$. Using a similar argument, we can prove $\|\pi^{(\ell)}\mathbf{1} - r\|_1 \geq \|(\pi^{(\ell+\frac{1}{2})})^\mathsf{T}\mathbf{1} - c\|_1$ for $\ell \geq 1$ and $\|\pi^{(0)}\mathbf{1} - r\|_1 + \|(\pi^{(0)})^\mathsf{T}\mathbf{1} - c\|_1 \geq \|(\pi^{(\frac{1}{2})})^\mathsf{T}\mathbf{1} - c\|_1$. The proof is completed. $\qquad\square$

**Lemma B.2** (Sufficient decrease of $\mathcal{L}_{\eta_k}(\cdot,\cdot)$ in $(\alpha,\beta)$). *Let $\{(\alpha^\ell,\beta^\ell)\}$ be the sequence generated by* (21) *and consider* $\{\pi^{(\ell)}\}$ *in* (22). *Then, for each $\ell \geq 0$, we have*

$$
\begin{aligned}
&\mathcal{L}_{\eta_k}(\alpha^{(\ell+1)}, \beta^{(\ell+1)}, U^{t+1}, \pi^k) \\
&\leq \mathcal{L}_{\eta_k}(\alpha^{(\ell)}, \beta^{(\ell)}, U^{t+1}, \pi^k) - \frac{\eta_k}{2}\left(\|\pi^{(\ell)}\mathbf{1} - r\|_1^2 + \|\pi^{(\ell+1)}\mathbf{1} - r\|_1^2\right).
\end{aligned}
\tag{47}
$$

*Proof.* By [29, Lemma 4.11], for $\ell \geq 0$, we have

$$
\mathcal{L}_{\eta_k}(\alpha^{(\ell+1)}, \beta^{(\ell+1)}, U^{t+1}, \pi^k) \leq \mathcal{L}_{\eta_k}(\alpha^{(\ell+1)}, \beta^{(\ell)}, U^{t+1}, \pi^k) - \frac{\eta_k}{2}\|(\pi^{(\ell+\frac{1}{2})})^\mathsf{T}\mathbf{1} - c\|_1^2.
$$

By Lemma B.1, for $\ell \geq 0$, we further have

$$
\mathcal{L}_{\eta_k}(\alpha^{(\ell+1)}, \beta^{(\ell+1)}, U^{t+1}, \pi^k) \leq \mathcal{L}_{\eta_k}(\alpha^{(\ell+1)}, \beta^{(\ell)}, U^{t+1}, \pi^k) - \frac{\eta_k}{2}\|(\pi^{(\ell+1)})\mathbf{1} - r\|_1^2. \tag{48}
$$

On the other hand, we have

$$
\begin{aligned}
&\mathcal{L}_{\eta_k}(\alpha^{(\ell+1)}, \beta^{(\ell)}, U^{t+1}, \pi^k) - \mathcal{L}_{\eta_k}(\alpha^{(\ell)}, \beta^{(\ell)}, U^{t+1}, \pi^k) = \langle r, \alpha^{(\ell+1)} - \alpha^{(\ell)}\rangle \\
&= -\eta_k\langle r, \log r - \log(\pi^{(\ell)}\mathbf{1})\rangle = -\eta_k\mathsf{KL}(r\|\pi^{(\ell)}\mathbf{1}) \leq -\frac{\eta_k}{2}\|\pi^{(\ell)}\mathbf{1} - r\|_1^2,
\end{aligned}
\tag{49}
$$

where the first equality uses the first equation in (21), the second equality comes from the second equation in (22), and the last inequality is due to Pinsker's inequality [33]. Here, for $p, q \in \Delta^n$, the Kullbbback-Leibler divergence between $p$ and $q$ is $\mathsf{KL}(p\|q) = \sum_i p_i \log(p_i/q_i)$. Combining (48) and (49), we obtain (47). The proof is completed. $\qquad\square$

We need to use the following elementary result.

**Lemma B.3.** *Let $\vartheta_1$ and $\vartheta_2$ be two given positive constants. Consider two sequences $\{a_\ell\}, \{b_\ell\} \subseteq \mathbb{R}_+$ with $\ell \geq 0$. If they obey:*

$$
a_\ell - a_{\ell+1} \geq \vartheta_1(b_\ell^2 + b_{\ell+1}^2), \tag{50a}
$$
$$
a_\ell \leq \vartheta_2 b_\ell \tag{50b}
$$

*for all $\ell \geq 1$, then we have $b_1 \leq \vartheta_2/\vartheta_1$ and*

$$
\min_{1 \leq i \leq \ell+1} b_i \leq \frac{\vartheta_2}{\vartheta_1} \cdot \frac{1}{\ell + \sqrt{2} - 1}, \qquad \forall\, \ell \geq 1.
$$

*Proof.* Applying (50) on $\ell = 1$, it is easy to see that $\vartheta_2 b_1 \geq a_1 \geq \vartheta_1 b_1^2$. Therefore, we have $b_1 \leq \vartheta_2/\vartheta_1$ and $a_1 \leq \vartheta_2^2/\vartheta_1$. For $\ell \geq 1$, by (50a), we have

$$
\frac{1}{a_{\ell+1}} - \frac{1}{a_\ell} = \frac{a_\ell - a_{\ell+1}}{a_l a_{\ell+1}} \geq \frac{\vartheta_1(b_\ell^2 + b_{\ell+1}^2)}{\vartheta_2^2 b_\ell b_{\ell+1}} \geq \frac{2\vartheta_1}{\vartheta_2^2},
$$

which, together with $a_1 \leq \vartheta_2^2/\vartheta_1$, implies

$$
a_{\ell+1} \leq \frac{\vartheta_2^2}{\vartheta_1} \cdot \frac{1}{2(\ell+1) - 1}, \qquad \forall\, \ell \geq 0.
$$

Let $\lfloor \ell/2 \rfloor$ be the largest integer smaller than $\ell/2$. For $\ell \geq 1$, summing (50a) over $i = \lfloor \ell/2 \rfloor + 1, \ldots, \ell$, we have

$$
2(\ell - \lfloor \ell/2 \rfloor)\min_{1 \leq i \leq l+1} b_i^2 \leq \sum_{i=\lfloor \ell/2 \rfloor + 1}^{\ell}(b_i^2 + b_{i+1}^2) \leq \frac{a_{\lfloor \ell/2 \rfloor + 1}}{\vartheta_1} \leq \frac{\vartheta_2^2}{\vartheta_1^2} \cdot \frac{1}{2(\lfloor \ell/2 \rfloor + 1) - 1},
$$

which immediately implies $\min_{1 \leq i \leq l+1} b_i^2 \leq \frac{\vartheta_2^2/\vartheta_1^2}{\ell(\ell+1)}$. With the fact that $\ell(\ell+1) \geq (\ell + \sqrt{2} - 1)^2$ for $\ell \geq 1$ and $b_1 \leq \vartheta_2/\vartheta_1$, we thus complete the proof. $\qquad\square$

With the help of the above lemmas, we can provide a new analysis of the iteration complexity of the classical Sinkhorn iteration based on the decreasing properties developed in Lemma B.1 and B.2. It differs from the proof technique in [19], wherein a switching strategy is adopted to establish the complexity.

**Lemma B.4.** *The total number of Sinkhorn iterations to find a point $\pi^{(\ell+1)}$ satisfying (23) is at most $\lceil 2R^{t+1}/(\eta_k\theta_{t+1}) + 2 - \sqrt{2}\rceil$, where*

$$R^{t+1} = \|C(U^{t+1})\|_{\mathrm{var}} + \eta_k\Psi, \tag{51}$$

*where $\Psi = \|\log\pi^k\|_{\mathrm{var}} + \max\{\|\log r\|_{\mathrm{var}}, \|\log c\|_{\mathrm{var}}\}$. Here, $\lceil a\rceil$ is the smallest nonnegative integer larger than $a \in \mathbb{R}^n$.*

*Proof.* Let $(\alpha^*_{U^{t+1}}, \beta^*_{U^{t+1}}) \in \mathrm{argmin}_{\alpha\in\mathbb{R}^n,\beta\in\mathbb{R}^n}\,\mathcal{L}_{\eta_k}(\alpha,\beta,U^{t+1},\pi^k)$. By refining the proof in [19] and [36], we can prove

$$\max\{\|\alpha^{(\ell+1)}\|_{\mathrm{var}}, \|\alpha^*_{U^{t+1}}\|_{\mathrm{var}}, \|\beta^{(\ell+1)}\|_{\mathrm{var}}, \|\beta^*_{U^{t+1}}\|_{\mathrm{var}}\} \le R^{t+1}, \quad \forall\,\ell \ge 0. \tag{52}$$

Since $\nabla_\alpha\mathcal{L}_{\eta_k}(\alpha^{(\ell+1)}, \beta^{(\ell+1)}, U^{t+1}, \pi^k) = r - \pi^{(\ell+1)}\mathbf{1}$,

$$\nabla_\beta\mathcal{L}_{\eta_k}(\alpha^{(\ell+1)}, \beta^{(\ell+1)}, U^{t+1}, \pi^k) = c - (\pi^{(\ell+1)})^\mathsf{T}\mathbf{1} = 0,$$

and $\mathcal{L}_{\eta_k}(\alpha,\beta,U^{t+1},\pi^k)$ is jointly convex with respect to $\alpha$ and $\beta$, we have

$$\mathcal{L}_{\eta_k}(\alpha^{(\ell+1)}, \beta^{(\ell+1)}, U^{t+1}, \pi^k) - \mathcal{L}_{\eta_k}(\alpha^*_{U^{t+1}}, \beta^*_{U^{t+1}}, U^{t+1}, \pi^k) \le \langle\pi^{(\ell+1)}\mathbf{1} - r, \alpha^*_{U^{t+1}} - \alpha^{(\ell+1)}\rangle. \tag{53}$$

Given $x \in \mathbb{R}^n$, let $x_{\mathrm{m}} = \|x\|_{\mathrm{var}}/2 + x_{\min}$, it holds that $\|x - x_{\mathrm{m}}\mathbf{1}\|_\infty = \|x\|_{\mathrm{var}}/2$. For $y \in \mathbb{R}^n$ with $\langle y, \mathbf{1}\rangle = 0$, we further know that

$$\langle y, x\rangle = \langle y, x - x_{\mathrm{m}}\mathbf{1}\rangle \le \|y\|_1\|x - x_{\mathrm{m}}\mathbf{1}\|_\infty = \frac{\|x\|_{\mathrm{var}}}{2}\|y\|_1.$$

Applying this assertion with $y = \pi^{(\ell+1)}\mathbf{1} - r$, $x = \alpha^*_{U^{t+1}}$ or $x = \alpha^{(\ell+1)}$, we obtain from $\|\pi^{(\ell+1)}\|_1 = \|r\|_1 = 1$, (52), and (53) that

$$\mathcal{L}_{\eta_k}(\alpha^{(\ell+1)}, \beta^{(\ell+1)}, U^{t+1}, \pi^k) - \mathcal{L}_{\eta_k}(\alpha^*_{U^{t+1}}, \beta^*_{U^{t+1}}, U^{t+1}, \pi^k) \le R^{t+1}\|\pi^{(\ell+1)}\mathbf{1} - r\|_1, \quad \forall\,\ell \ge 0. \tag{54}$$

Applying Lemma B.3 with $a_\ell = \mathcal{L}_{\eta_k}(\alpha^{(\ell)}, \beta^{(\ell)}, U^{t+1}, \pi^k) - \mathcal{L}_{\eta_k}(\alpha^*_{U^{t+1}}, \beta^*_{U^{t+1}}, U^{t+1}, \pi^k)$ and $b_\ell = \|\pi^{(\ell)}\mathbf{1} - r\|_1$ and using (46), (47), and (54), we have $\|\pi^{(1)}\mathbf{1} - r\|_1 \le 2R^{t+1}/\eta_k$ and $\|\pi^{(\ell+1)}\mathbf{1} - r\|_1 \le 2R^{t+1}/(\eta_k(\ell + \sqrt{2} - 1))$ for all $\ell \ge 1$. Letting $2R^{t+1}/(\eta_k(\ell + \sqrt{2} - 1)) \le \theta_{t+1}$, we can complete the proof of Lemma B.4. $\qquad\square$

We now can establish the convergence result of iRBBS, namely, Algorithm 2.

**Theorem B.5.** *Let $\{\mathbf{x}^t\}$ be the sequence generated by Algorithm 2. If $\epsilon_1 = \epsilon_2 = 0$, we have $\mathsf{e}^1_{\eta_k}(\mathbf{x}^t, \pi^k) \to 0$ and $\mathsf{e}^2_{\eta_k}(\mathbf{x}^t, \pi^k) \to 0$. If $\epsilon_1 > 0$ and $\epsilon_2 > 0$, then Algorithm 2 stops within at most $\lceil\Upsilon\max\{\epsilon_1^{-2}, \epsilon_2^{-2}\}\rceil$ iterations for any $\theta_{t+1} \ge 0$, where $\Upsilon$ is a constant defined in (64).*

*Proof.* Given $\alpha \in \mathbb{R}^n$ and $\beta \in \mathbb{R}^n$, by [29, Lemma 4.8] for any $\iota \in [0, 1]$ and $U_1, U_2 \in \mathcal{U}$,

$$\begin{aligned}&\|\nabla_U\mathcal{L}_{\eta_k}(\alpha, \beta, U_2, \pi^k) - \nabla_U\mathcal{L}_{\eta_k}(\alpha, \beta, \iota U_1 + (1 - \iota)U_2, \pi^k)\|_\mathsf{F}\\&\le 2\left(\|C\|_\infty + 2\|C\|_\infty^2/\eta\right)\iota\|U_1 - U_2\|_\mathsf{F}.\end{aligned} \tag{55}$$

From [38, 10], we know that the retraction on the Stiefel manifold has the following nice properties, namely, there exist positive constants $L_1$ and $L_2$ such that

$$\|\mathsf{Retr}_U(\xi) - U\|_\mathsf{F} \le L_1\|\xi\|_\mathsf{F} \quad\text{and}\quad \|\mathsf{Retr}_U(\xi) - (U + \xi)\|_\mathsf{F} \le L_2\|\xi\|_\mathsf{F}^2 \tag{56}$$

hold for all $\xi \in \mathrm{T}_U\mathcal{U}$ and $U \in \mathcal{U}$. With (55) and (56), following the proof of Lemma 3 in [10], for each $U \in \mathcal{U}$ and $\xi \in \mathrm{T}_U\mathcal{U}$, we have

$$\mathcal{L}_{\eta_k}\left(\alpha, \beta, \mathsf{Retr}_U(\xi), \pi^k\right) \le \mathcal{L}_{\eta_k}(\alpha, \beta, U, \pi^k) + \left\langle\mathrm{grad}_U\mathcal{L}_{\eta_k}(\mathbf{x}, \pi^k), \xi\right\rangle + \frac{L}{2}\|\xi\|_\mathsf{F}^2, \tag{57}$$

where

$$L = 2(L_1^2 + L_2)\|C\|_\infty + 4L_1^2\|C\|_\infty^2/\eta_k. \tag{58}$$

Applying (57) with $\alpha = \alpha^t, \beta = \beta^t, U = U^t$, and $\xi = -\tau_t \operatorname{grad}_U \mathcal{L}_{\eta_k}(\mathbf{x}^t, \pi^k) = -\tau_t \xi^t$, with (19a), we have

$$\mathcal{L}_{\eta_k}(\alpha^t, \beta^t, U^{t+1}, \pi^k) \leq \mathcal{L}_{\eta_k}(\mathbf{x}^t, \pi^k) - \tau_t \left(1 - \frac{L\tau_t}{2}\right) \|\xi^t\|_\mathsf{F}^2. \tag{59}$$

In addition, by the fact that $\alpha^{t+1} = \alpha^{(\ell+1)}, \beta^{t+1} = \beta^{(\ell+1)}$, and $\mathrm{e}_{\eta_k}^2(\mathbf{x}^{t+1}, \pi^k) = \|\pi^{(\ell+1)}\mathbf{1} - r\|_1$ for some $\ell$ and $\alpha^{(0)} = \alpha^t, \beta^{(0)} = \beta^t$, we have from (47) that

$$\mathcal{L}_{\eta_k}(\mathbf{x}^{t+1}, \pi^k) \leq \mathcal{L}_{\eta_k}(\alpha^t, \beta^t, U^{t+1}, \pi^k) - \frac{\eta_k}{2}(\mathrm{e}_{\eta_k}^2(\mathbf{x}^{t+1}, \pi^k))^2. \tag{60}$$

Recalling (24), it follows from (59) and (60) that

$$E_\rho(\mathbf{x}^{t+1}) \leq E_\rho(\mathbf{x}^t) - \tau_t\left(1 - \frac{L\tau_t}{2}\right)\|\xi^t\|_\mathsf{F}^2 - \left(\frac{\eta_k}{2} - \rho\right)(\mathrm{e}_{\eta_k}^2(\mathbf{x}^{t+1}, \pi^k))^2 \tag{61}$$

holds with $\rho \in [0, \eta_k/2)$.

Recalling that $\tau_t$ takes the form $\tau_t = \tau_t^{(0)}\sigma^s$ for some nonnegative integer $s$, with (26), we know from (61) that if $s = \lceil \log \frac{2(1-\delta_1)}{\tau_{\max} L} / \log \sigma \rceil$, there must hold that

$$E_\rho(\mathbf{x}^{t+1}) \leq E_\rho(\mathbf{x}^t) - \delta_1 \tau_t \|\xi^t\|_\mathsf{F}^2 - \left(\frac{\eta_k}{2} - \rho\right)(\mathrm{e}_{\eta_k}^2(\mathbf{x}^{t+1}, \pi^k))^2.$$

Using similar arguments as [54, Lemma 1.1], this also shows that the backtracking line search in Algorithm 2 terminates in at most $\lceil \log \frac{2(1-\delta_1)}{\tau_{\max} L} / \log \sigma \rceil$ trials and $E_t^r \geq E_\rho(\mathbf{x}^t)$. Moreover, since $\tau_t^{(0)} \geq \tau_{\min}$ from (26), we further know that $\tau_t = \tau_t^{(0)}\sigma^s \geq \underline{\tau} := \frac{2\sigma(1-\delta_1)\tau_{\min}}{\tau_{\max} L}$. Therefore, we know that the nonmonotone linesearch condition (25) is satisfied for some $\tau_t \geq \underline{\tau}$.

Let $\mathbf{x}^* = (\alpha^*, \beta^*, U^*)$ be the minimizer of problem (11). Using a similar argument as [28, Lemma 4.7], by (34) and $-\log\pi_{ij}^k \leq \|\log\pi^k\|_\infty$, we have

$$\mathcal{L}_{\eta_k}(\mathbf{x}^*, \pi^k) \geq -\|C\|_\infty - \eta_k\|\log\pi^k\|_\infty. \tag{62}$$

Let $\omega_t = E_t^r - E_\rho(\mathbf{x}^t)$. By (62) and using the argument in [44, Theorem 5], we know that such $\omega_t \geq 0$ satisfies

$$\sum_{t=0}^{+\infty} \omega_t \leq \frac{\gamma}{1-\gamma}(E_{\eta_k/2}(\mathbf{x}^0) + \|C\|_\infty + \eta_k\|\log\pi^k\|_\infty) < +\infty.$$

By (24), the fact that (25) holds with $\tau_t \geq \underline{\tau}$, and $\|\xi^t\|_\mathsf{F} = \mathrm{e}_{\eta_k}^1(\mathbf{x}^t, \pi^k)$, we have

$$\begin{aligned}
&\delta_1\underline{\tau}(\mathrm{e}_{\eta_k}^1(\mathbf{x}^t, \pi^k))^2 + \left(\frac{\eta_k}{2} - \rho\right)(\mathrm{e}_{\eta_k}^2(\mathbf{x}^{t+1}, \pi^k))^2 \\
&\leq \mathcal{L}_{\eta_k}(\mathbf{x}^t, \pi^k) - \mathcal{L}_{\eta_k}(\mathbf{x}^{t+1}, \pi^k) + \rho((\mathrm{e}_{\eta_k}^2(\mathbf{x}^t, \pi^k))^2 - (\mathrm{e}_{\eta_k}^2(\mathbf{x}^{t+1}, \pi^k))^2) + \omega_t.
\end{aligned} \tag{63}$$

Summing (63) over $t = 0, \ldots, T$ and adding the term $(\eta_k/2 - \rho)(\mathrm{e}_{\eta_k}^2(\mathbf{x}^0))^2$ on both sides of the obtained equation, and then combining them with $\mathcal{L}_{\eta_k}(\mathbf{x}^{T+1}, \pi^k) \geq \mathcal{L}_{\eta_k}(\mathbf{x}^*, \pi^k)$ and (62), we have

$$\sum_{t=0}^T \left(\delta_1\underline{\tau}(\mathrm{e}_{\eta_k}^1(\mathbf{x}^t, \pi^k))^2 + \left(\frac{\eta_k}{2} - \rho\right)(\mathrm{e}_{\eta_k}^2(\mathbf{x}^t, \pi^k))^2\right)$$

$$\leq \widetilde{\Upsilon} := \frac{1}{1-\gamma}\left(E_{\eta_k/2}(\mathbf{x}^0) + \|C\|_\infty + \eta_k\|\log\pi^k\|_\infty\right)$$

for any $T \geq 1$. This further implies

$$\sum_{t=0}^T (\mathrm{e}_{\eta_k}^1(\mathbf{x}^t, \pi^k))^2 + (\mathrm{e}_{\eta_k}^2(\mathbf{x}^t, \pi^k))^2 \leq \Upsilon := \frac{\widetilde{\Upsilon}}{\min\{\delta_1\underline{\tau}, \eta_k/2 - \rho\}}. \tag{64}$$

Hence, we have $\mathrm{e}_{\eta_k}^1(\mathbf{x}^t, \pi^k) \to 0$ and $\mathrm{e}_{\eta_k}^2(\mathbf{x}^t, \pi^k) \to 0$ as $t \to \infty$.

Suppose $e^1_{\eta_k}(\mathbf{x}^t, \pi^k) \leq \epsilon_1$ and $e^2_{\eta_k}(\mathbf{x}^t, \pi^k) \leq \epsilon_2$ are fulfilled for $t = \overline{T}$ but not fulfilled for all $t < \overline{T}$. Then there hold that $e^1_{\eta_k}(\mathbf{x}^{\overline{T}}, \pi^k) \leq \epsilon_1$, $e^2_{\eta_k}(\mathbf{x}^{\overline{T}}, \pi^k) \leq \epsilon_2$, and $(e^1_{\eta_k}(\mathbf{x}^t, \pi^k))^2 + (e^2_{\eta_k}(\mathbf{x}^t, \pi^k))^2 > \min\{\epsilon_1^2, \epsilon_2^2\}$ for all $t < \overline{T}$. Setting $T$ in (64) as $\overline{T}$ yields $\overline{T}\min\{\epsilon_1^2, \epsilon_2^2\} \leq \Upsilon$, which completes the proof. □

We are now ready to complete the proof of Theorem 3.2.

*Proof of Theorem 3.2.* Given $\epsilon_1 > 0$ and $\epsilon_2 > 0$, by Theorem B.5, we know that Algorithm 2 will return a point $\mathbf{x}^T$ with $T \leq \lceil \Upsilon \max\{\epsilon_1^{-2}, \epsilon_2^{-2}\}\rceil$ satisfying satisfying $e^1_{\eta_k}(\mathbf{x}^t, \pi^k) \leq \epsilon_1$ and $e^2_{\eta_k}(\mathbf{x}^t, \pi^k) \leq \epsilon_2$. Let $\hat{\pi}^T := \texttt{Round}(\phi_{\eta_k}(\mathbf{x}^T, \pi^k), \Pi(r, c))$, where "Round" is the rounding procedure given in [3, Algorithm 2]. By the second equation in (22) and $\mathbf{x}^T = (\alpha^T, \beta^T, U^T)$ with $\alpha^T = \alpha^{(\ell+1)}, \beta^T = \beta^{(\ell+1)}$ for some $\ell$, we know that $\|\zeta_{\eta_k}(\mathbf{x}^T, \pi^k)\|_1 = 1$. By $\|C(U)\|_\infty \leq \|C\|_\infty$ for all $U \in \mathcal{U}$, (58), (52), (51), we have $\|\alpha^T\|_{\text{var}} + \|\beta^T\|_{\text{var}} + \|C\|_\infty \leq 3\|C\|_\infty + 2\eta_k\Psi$. Therefore, we obtain from (13) that

$$\|\text{Proj}_{\text{T}_U\mathcal{U}}(-2V_{\hat{\pi}^T}U^T)\|_{\mathsf{F}} \leq \epsilon_1 + 2\|C\|_\infty\epsilon_2,$$
$$\langle \hat{\pi}^T, Z(\mathbf{x}^T)\rangle \leq (2\log n + \|\log \pi^k\|_{\text{var}})\eta_k + (3\|C\|_\infty + 2\eta_k\Psi)\epsilon_2.$$

With the choices of $\eta_k$, $\epsilon_1$ and $\epsilon_2$ in Theorem 3.2, we can immediately know from (13) that such $(\mathbf{x}^T, -\varphi(\mathbf{x}^T)_{\min})$ is an $(\epsilon'_1, \epsilon'_2)$-stationary point of problem (6). With the expression of $\Upsilon$ in (64), Lemma B.4 and Remark 3.5, we can complete the proof. Note that the $\mathcal{O}(\cdot)$ in this theorem hides the constants related to $E_{\eta_k/2}(\mathbf{x}^0)$, $\Psi$, $L_1$, $L_2$, $\log n$, and $\|C\|_\infty$. □

# C   Experimental Settings

We implemented all methods in MATLAB 2021b and performed all the experiments on a MacBook Pro with a 2.3GHz 8-core Intel Core i9. We follow the ways in [34] to generate the following real dataset.

**Datasets C.1.** *We consider six Shakespeare operas. Each script is preprocessed according to the way in [34] and corresponds to a matrix over $X \in \mathbb{R}^{300 \times n_X}$. The values $n_X$ of H5, H, JC, MV, O, and RJ are 1303, 1481, 910, 1008, 1148, and 1141, respectively. The weight vector $r$ or $c$ is taken as $\mathbf{1}_{n_X}/n_X$. We set $k = 2$.*

*Initial points and retraction.* We choose $\alpha^{-1} = \mathbf{0}$, $\beta^{-1} = \mathbf{0}$ for iRBBS and $\alpha^0 = \mathbf{0}, \beta^0 = \mathbf{0}$ for R(A)BCD. As for $U^0$, we choose $U^0 \in \text{argmax}_{U \in \mathcal{U}}\langle V_{\pi^0}, UU^\mathsf{T}\rangle$ with $\pi^0 \in \Pi(r, c)$ for all methods. Here, $\pi^0$ is formulated by firstly generating a matrix $\tilde{\pi}^0$ with each entry randomly drawn from the standard uniform distribution on $(0, 1)$ and then rounding $\tilde{\pi}^0/\|\tilde{\pi}^0\|_1$ to $\pi^0$ via [3, Algorithm 2]. The retraction operator in all the above methods is taken as the QR-retraction $\text{Retr}^{\text{qr}}_U(\xi) = \text{qf}(X + \xi)$, where $\text{qf}(X + \xi) \in \mathcal{U}$ satisfying $X + \xi = \text{qf}(X + \xi)\text{upp}(X + \xi)$ with $\text{upp}(X + \xi)$ being an upper triangular $k$-by-$k$ matrix with positive diagonal elements.

*Parameters of iRBBS in Section 4.1.* We choose $\tau^{\min} = 10^{-10}/L$, $\tau^{\max} = 10^{10}/L$, $\gamma = 0.85$, $\tau_0 = 10^{-3}$, $\sigma = 1/2$, $\delta_1 = 10^{-4}$, and $\rho = 0.49\eta_k$.

*Parameters of R(A)BCD in Section 4.1.* As stated in Remark 4.1 in [28], to achieve the best performance of R(A)BCD, one has to spend some efforts in tuning the stepsizes. Here, we adopt the stepsizes used therein (with a slight modification, marked in *italic* type, to have better performance for some cases). We choose $\tau_{\texttt{RBCD}} = 0.09$ if the instance is X/RJ and $\tau_{\texttt{RBCD}} = 0.1$ otherwise; $\tau_{\texttt{RABCD}} = 0.0015$ if the instance is X/RJ with X $\neq$ H, $\tau_{\texttt{RABCD}} = 0.001$ if the instance is H/RJ and $\tau_{\texttt{RABCD}} = 0.0025$ otherwise. Moreover, we stop R(A)BCD when $e^1_{\eta_k}(\alpha^t, \beta^{t-1}, U^t) \leq \epsilon_1$ and $e^2_{\eta_k}(\alpha^t, \beta^{t-1}, U^t) \leq \epsilon_2$ or the maximum iteration number reaches 5000.

*Parameters of REALM in Section 4.2.* Once $\eta_k$ becomes $\eta_{\min}$, we set the corresponding $\epsilon_{k,1} = \epsilon_1$ and $\epsilon_{k,2} = \epsilon_2$ and stop the algorithm if $e^1_{\eta_k}(\mathbf{x}^k, \pi^k) \leq \epsilon_1$ and $e^2_{\eta_k}(\mathbf{x}^k, \pi^k) \leq \epsilon_2$. We choose $\epsilon_{1,2} = 10^{-1}\max\{\|r\|_\infty, \|c\|_\infty\}$ and $\epsilon_{1,1} = 2\|C\|_\infty\epsilon_{1,2}$.

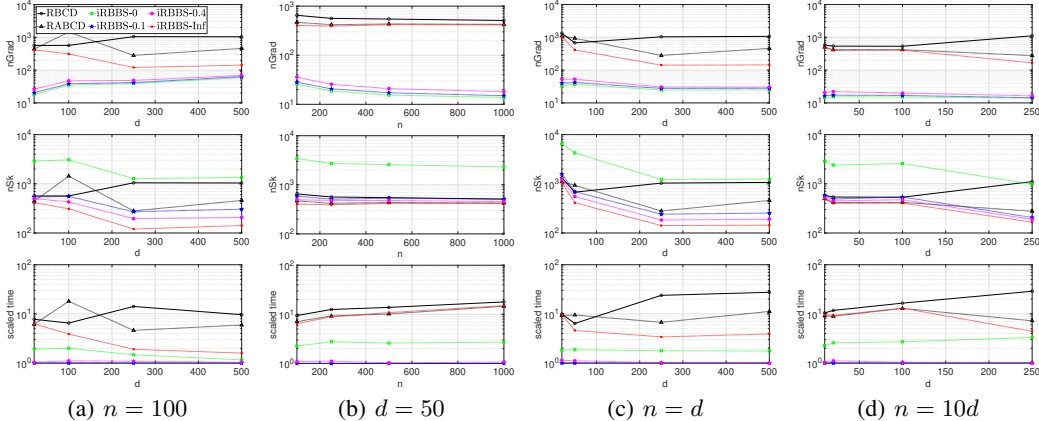

(a) $n = 100$    (b) $d = 50$    (c) $n = d$    (d) $n = 10d$

Figure 2: Averaged results for Dataset D.1. The scaled time means the time of each method over the minimum time of all methods. For (a), $d \in \{20, 100, 250, 500\}$; for (b), $n \in \{100, 250, 500, 1000\}$; for (c), $d \in \{20, 50, 250, 500\}$; for (d), $d \in \{10, 20, 100, 250\}$.

## D   Additional Numerical Results

In this section, we present additional numerical results to evaluate the performance of our proposed methods to compute the PRW distance (1). We follow the ways in [34, 28, 41] to generate the synthetic and real datasets.

**Datasets D.1** (Synthetic dataset: Fragmented hypercube [41]). *Define a map $P(x) = x + 2\text{sign}(x) \odot \sum_{k=1}^{k^*} e_k$, where $\text{sign}(\cdot)$ is taken elementwise, $k^* \in \{1, \dots, d\}$ and $\{e_1, \dots, e_{k^*}\}$ is the canonical basis of $\mathbb{R}^d$. Let $\mu = \mathcal{U}([-1, 1]^d)$ be the uniform distribution over an hypercube and $\nu = P_{\#}(\mu)$ be the pushforward of $\mu$ under the map $P$. We set $k = k^* = 2$ and take both the weight vectors $r$ and $c$ as $\mathbf{1}/n$.*

**Datasets D.2** (Real datasets: digits from MNIST datasets [34]). *For each digit $0, 1, \dots, 9$, we extract the 128-dimensional features from a pre-trained convolutional neural network. Each digit corresponds to a matrix $X \in \mathbb{R}^{128 \times n_X}$. The values $n_X$ of $0, 1, \dots, 9$ are 980, 1135, 1032, 1010, 982, 892, 958, 1028, 974, and 1009, respectively. The weight vector $r$ or $c$ is taken as $\mathbf{1}_{n_X}/n_X$. We choose $k = 2$.*

### D.1   Comparison with R(A)BCD on Solving Subproblem (11)

We follow the same settings of $\eta$ in [28]. For Dataset D.1, we choose $\eta = 0.2$ when $d < 250$ and $\eta = 0.5$ otherwise. For Dataset D.2, we set $\eta = 8$.

*Parameters of R(A)BCD in this section.* For Dataset D.1, we choose $\tau_{\text{RBCD}} = \tau_{\text{RABCD}} = 0.001$ as done in [28]. For Dataset D.2, we choose $\tau_{\text{RBCD}} = 0.004$, and we do not test RABCD for this dataset since the well-chosen stepsize is not provided in [28].

For Dataset D.1, we randomly generate 10 instances for each $(n, d)$ pair, each equipped with 5 randomly generated initial points. The comparison results for Dataset D.1 are plotted as Figure 2. Note that the values $\widehat{\mathcal{P}}_k^2$ returned by different methods are almost the same for this dataset. Therefore, we do not report the values in the figure. From the figure, we can draw the following observations. (i) Among iRBBS methods, iRBBS with smaller $\theta$ always has less nGrad but more nSk. This means that computing $\text{grad}\, q(U^t)$ in a relatively high precision can help to reduce the whole iteration number of updating $U$. In particular, iRBBS-0, which computes (almost) the exact $\text{grad}\, q(U^t)$, takes the least nGrad. Because the complexity of one Sinkhorn iteration is much less than that of updating $U$ (see Remark 3.5), iRBBS with a moderate $\theta$ generally achieves the best overall performance. The last rows of Figure 2 show that iRBBS-0.1 is almost the fastest among all iRBBS methods. (ii) Our iRBBS-$\theta$ is better than R(A)BCD in terms that the former always have smaller nGrad and nSk and are always faster. Particularly, for Dataset D.1, iRBBS-0.1 is always more than 5x faster than RABCD and is about more than 10x faster than RBCD. For the largest instance $d = 250$, $n = 2500$, iRBBS-0.1 (1.6s) is more than 7.2x faster than RABCD (11.1s) and about 28.6x faster than RBCD (44.4s). It

Table 3: The average results of 20 runs for Dataset D.2. Here, "a", "c", "d", and "e" stand for RBCD, iRBBS-inf, iRBBS-0.1, and iRBBS-0, respectively. For RBCD and iRBBS-inf, nGrad is equal to nSk.

| data | $10^{-3} \times \widehat{\mathcal{P}}_k^2$ | | | | nGrad/nSk | | | | time | | | |
|---|---|---|---|---|---|---|---|---|---|---|---|---|
| | a | c | d | e | a | c | d | e | a | c | d | e |
| D0/D1 | 0.974647 | 0.974647 | 0.974647 | **0.974647** | 519 | 132 | 59/506 | 46/1062 | 2.7 | 0.7 | **0.4** | 0.4 |
| D0/D2 | 0.794227 | **0.794227** | 0.794227 | 0.794227 | 988 | 187 | 72/943 | 74/2655 | 4.8 | 1.0 | **0.5** | 0.6 |
| D0/D3 | 1.202136 | 1.202136 | 1.202136 | **1.202136** | 554 | 177 | 64/934 | 44/1386 | 2.6 | 0.8 | 0.4 | **0.4** |
| D0/D4 | 1.220464 | 1.220464 | **1.230833** | 1.230833 | 912 | 243 | 98/998 | 81/2947 | 4.1 | 1.1 | **0.5** | 0.6 |
| D0/D5 | 1.026643 | **1.027569** | 1.026643 | 1.026643 | 1178 | 252 | 89/1402 | 96/4355 | 4.8 | 1.1 | **0.5** | 0.8 |
| D0/D6 | 0.802711 | 0.802711 | 0.802711 | **0.802711** | 919 | 186 | 66/1052 | 54/2387 | 4.0 | 0.8 | **0.4** | 0.5 |
| D0/D7 | 0.855776 | 0.855776 | **0.855776** | 0.855776 | 601 | 163 | 63/747 | 56/1727 | 2.7 | 0.8 | **0.4** | 0.4 |
| D0/D8 | 1.052850 | **1.052850** | 1.052850 | 1.052850 | 709 | 185 | 84/1330 | 72/3302 | 3.1 | 0.8 | **0.5** | 0.6 |
| D0/D9 | 1.084633 | **1.084633** | 1.084633 | 1.084633 | 698 | 179 | 68/1108 | 55/2436 | 3.3 | 0.8 | **0.4** | 0.5 |
| D1/D2 | 0.664666 | **0.664666** | 0.664666 | 0.664666 | 757 | 114 | 55/638 | 47/1511 | 4.1 | 0.6 | **0.4** | 0.5 |
| D1/D3 | 0.861031 | **0.861031** | 0.861031 | 0.861031 | 1649 | 230 | 105/982 | 71/2065 | 8.8 | 1.3 | **0.7** | 0.8 |
| D1/D4 | 0.666639 | 0.666639 | 0.666639 | **0.666639** | 982 | 153 | 65/505 | 54/1282 | 5.1 | 0.8 | **0.4** | 0.5 |
| D1/D5 | **0.839978** | 0.839978 | 0.839978 | 0.839978 | 3436 | 540 | 184/1503 | 157/3844 | 15.9 | 2.4 | **1.0** | 1.1 |
| D1/D6 | 0.795720 | 0.795720 | **0.795720** | 0.795368 | 2919 | 331 | 128/1095 | 119/3354 | 15.0 | 1.7 | **0.8** | 1.1 |
| D1/D7 | **0.572325** | 0.572325 | 0.572325 | 0.572325 | 961 | 157 | 65/525 | 56/1253 | 5.2 | 0.9 | **0.4** | 0.5 |
| D1/D8 | 0.878387 | **0.878387** | 0.878387 | 0.878310 | 5000 | 402 | 200/2001 | 181/5505 | 25.8 | 2.1 | **1.3** | 1.6 |
| D1/D9 | 0.853405 | 0.853405 | **0.853405** | 0.853405 | 951 | 168 | 69/581 | 58/1321 | 5.6 | 1.0 | **0.6** | 0.7 |
| D2/D3 | **0.718821** | 0.718821 | 0.718821 | 0.718821 | 2070 | 336 | 120/1733 | 100/4897 | 10.0 | 1.6 | **0.8** | 1.0 |
| D2/D4 | 1.068421 | 1.085370 | **1.085370** | 1.085370 | 5000 | 4602 | 550/8093 | 382/16291 | 24.3 | 21.4 | **3.2** | 3.4 |
| D2/D5 | 1.076728 | 1.076728 | **1.076728** | 1.076728 | 1320 | 272 | 109/1910 | 89/3990 | 5.9 | 1.2 | **0.6** | 0.7 |
| D2/D6 | **0.899301** | 0.899301 | 0.899301 | 0.899301 | 1202 | 245 | 102/967 | 87/3205 | 5.8 | 1.2 | **0.6** | 0.7 |
| D2/D7 | 0.695023 | **0.695023** | 0.695023 | 0.695023 | 951 | 174 | 67/1156 | 63/3108 | 4.7 | 0.9 | **0.5** | 0.8 |
| D2/D8 | 0.671073 | 0.671073 | 0.671073 | **0.671073** | 1293 | 213 | 72/925 | 69/3547 | 6.2 | 1.0 | **0.4** | 0.7 |
| D2/D9 | 1.056729 | **1.056729** | 1.056729 | 1.056729 | 1985 | 284 | 120/1907 | 93/3753 | 9.4 | 1.3 | **0.7** | 0.8 |
| D3/D4 | 1.199367 | 1.200346 | **1.200346** | 1.200346 | 5000 | 312 | 76/762 | 69/2523 | 22.2 | 1.4 | **0.4** | 0.5 |
| D3/D5 | 0.581711 | **0.581711** | 0.581711 | 0.581711 | 1187 | 181 | 71/793 | 58/2430 | 4.8 | 0.8 | **0.4** | 0.5 |
| D3/D6 | 1.231675 | **1.231675** | 1.231675 | 1.231675 | 1028 | 212 | 82/772 | 66/2121 | 4.5 | 0.9 | **0.4** | 0.5 |
| D3/D7 | **0.724492** | 0.724492 | 0.724492 | 0.724492 | 1101 | 202 | 88/831 | 776/19184 | 5.2 | 0.9 | **0.5** | 5.3 |
| D3/D8 | 0.879795 | 0.879795 | **0.879795** | 0.879795 | 916 | 200 | 84/1113 | 77/3675 | 4.0 | 0.9 | **0.5** | 0.7 |
| D3/D9 | **0.830151** | 0.830151 | 0.830151 | 0.830151 | 1572 | 210 | 72/955 | 49/2193 | 7.2 | 1.0 | **0.4** | 0.4 |
| D4/D5 | 1.006255 | 1.006255 | **1.006255** | 1.006255 | 936 | 245 | 87/863 | 71/2748 | 3.9 | 1.0 | **0.4** | 0.5 |
| D4/D6 | **0.843616** | 0.843616 | 0.843616 | 0.843616 | 1312 | 245 | 96/1102 | 82/3463 | 5.9 | 1.1 | **0.5** | 0.7 |
| D4/D7 | 0.789906 | **0.789906** | 0.789906 | 0.789906 | 1351 | 269 | 96/912 | 76/2509 | 6.4 | 1.2 | **0.5** | 0.6 |
| D4/D8 | 1.094979 | 1.098234 | 1.098234 | **1.098234** | 5000 | 213 | 52/550 | 52/1930 | 21.9 | 0.9 | **0.3** | 0.4 |
| D4/D9 | 0.489215 | 0.489215 | **0.489215** | 0.489215 | 1997 | 281 | 70/1116 | 54/2610 | 9.1 | 1.3 | **0.4** | 0.5 |
| D5/D6 | **0.717009** | 0.717009 | 0.717009 | 0.717009 | 1335 | 220 | 86/891 | 69/2536 | 5.4 | 0.9 | **0.4** | 0.5 |
| D5/D7 | **0.912583** | 0.912583 | 0.912583 | 0.912583 | 899 | 242 | 72/652 | 71/2132 | 3.7 | 1.0 | **0.4** | 0.5 |
| D5/D8 | 0.715209 | 0.715209 | **0.715209** | 0.715209 | 1429 | 225 | 71/1060 | 50/2347 | 5.9 | 0.9 | **0.4** | 0.4 |
| D5/D9 | 0.775506 | 0.775506 | 0.775506 | **0.775506** | 1413 | 252 | 86/1216 | 64/3451 | 5.9 | 1.0 | **0.5** | 0.6 |
| D6/D7 | 1.109787 | **1.109787** | 1.109787 | 1.109787 | 826 | 195 | 66/398 | 62/1318 | 3.8 | 0.9 | **0.3** | 0.4 |
| D6/D8 | 0.917487 | 0.917487 | **0.917487** | 0.917487 | 725 | 186 | 57/721 | 57/2305 | 3.2 | 0.8 | **0.3** | 0.5 |
| D6/D9 | 1.105550 | **1.105550** | 1.105550 | 1.105550 | 1117 | 219 | 95/1038 | 79/3046 | 5.0 | 1.0 | **0.5** | 0.6 |
| D7/D8 | 1.078740 | **1.078740** | 1.078740 | 1.078740 | 757 | 180 | 68/796 | 59/2455 | 3.5 | 0.8 | **0.4** | 0.5 |
| D7/D9 | **0.607866** | 0.607866 | 0.607866 | 0.607866 | 1985 | 266 | 79/1026 | 72/3208 | 9.3 | 1.3 | **0.5** | 0.7 |
| D8/D9 | 0.867753 | 0.867753 | **0.867753** | 0.867753 | 745 | 180 | 57/713 | 50/2107 | 3.2 | 0.8 | **0.3** | 0.4 |
| AVG | 0.884689 | 0.885180 | **0.885390** | 0.885380 | 1560 | 326 | 95/1152 | 95/3366 | 7.28 | 1.52 | **0.57** | 0.78 |

should also be mentioned that for all 800 problem instances, there are 5/21 instances for which RBCD/RABCD meet the maximum iteration numbers and return solutions not satisfying the stopping criteria.

For each instance of Dataset D.2, we randomly generate 20 initial points. The comparison results for Dataset D.2 are reported in Table 3. From the table, we have the following observations: (i) as for the solution quality, iRBBS-0 and iRBBS-0.1 perform the best among all methods; (ii) among iRBBS methods, iRBBS-0.1 and iRBBS-0 take the least nGrad and iRBBS-inf takes the most nGrad

Table 4: The average results of REALM for Dataset D.1. In this table, "a" and "b" stand for REALM-$(0.02, 0)$ and REALM-$(0.055, 0.9)$, respectively.

| $n/d$ | $\widehat{\mathcal{P}}_k^2$ a | $\widehat{\mathcal{P}}_k^2$ b | nGrad a | nGrad b | $\mathrm{nSk_{exp}}/\mathrm{nSk_{log}}$ a | $\mathrm{nSk_{exp}}/\mathrm{nSk_{log}}$ b | time a | time b | iter a | iter b |
|---|---|---|---|---|---|---|---|---|---|---|
| 100/20 | **8.34297** | 8.34296 | 182 | 138 | 3208/8342 | 13258/0 | 1.1 | 0.1 | 0.0/7.0 | 8.0/14.0 |
| 100/100 | 9.17320 | **9.17340** | 287 | 241 | 3837/3354 | 11369/0 | 0.5 | 0.1 | 0.0/7.0 | 8.0/14.0 |
| 100/250 | 10.89561 | **10.89689** | 467 | 348 | 3200/8858 | 10550/279 | 1.3 | 0.2 | 0.0/7.0 | 8.0/14.1 |
| 100/500 | 13.31111 | **13.31524** | 680 | 469 | 2192/3782 | 9844/213 | 0.8 | 0.2 | 0.0/7.0 | 8.0/14.0 |
| 100/50 | 8.59993 | **8.60005** | 321 | 189 | 3945/13096 | 11700/0 | 1.8 | 0.1 | 0.0/7.0 | 8.0/14.0 |
| 250/50 | **8.23339** | 8.23336 | 150 | 156 | 2480/4168 | 10828/0 | 2.2 | 0.3 | 0.0/7.0 | 7.9/13.9 |
| 500/50 | **8.12987** | 8.12986 | 122 | 141 | 1944/4508 | 11578/0 | 7.2 | 0.8 | 0.0/7.0 | 8.0/14.0 |
| 1000/50 | **8.07096** | 8.07095 | 111 | 130 | 1716/5307 | 12344/0 | 30.8 | 2.3 | 0.0/7.0 | 8.0/14.0 |
| 20/20 | 9.30977 | **9.31036** | 591 | 221 | 4016/23861 | 11906/0 | 0.5 | 0.0 | 0.0/7.1 | 8.0/14.0 |
| 50/50 | 9.36270 | **9.36292** | 386 | 248 | 4656/9416 | 11819/0 | 0.5 | 0.1 | 0.0/7.0 | 7.9/13.9 |
| 250/250 | 9.17441 | **9.17476** | 282 | 257 | 2942/4136 | 11371/248 | 2.2 | 0.5 | 0.0/7.0 | 8.0/14.0 |
| 500/500 | 9.11498 | **9.11540** | 258 | 262 | 2391/4884 | 9927/873 | 7.5 | 2.2 | 0.0/7.0 | 7.6/13.6 |
| 100/10 | **8.16197** | 8.16193 | 405 | 173 | 4635/33570 | 65772/0 | 4.6 | 0.3 | 0.0/7.0 | 8.0/14.0 |
| 200/20 | **8.13310** | 8.13309 | 180 | 128 | 2861/10259 | 15007/0 | 3.8 | 0.3 | 0.0/7.0 | 8.0/14.0 |
| 1000/100 | **8.11874** | 8.11873 | 124 | 145 | 2096/5157 | 12153/141 | 30.5 | 3.1 | 0.0/7.0 | 8.0/14.0 |
| 2500/250 | **8.11640** | 8.11639 | 117 | 146 | 2416/5499 | 11482/624 | 373.8 | 103.3 | 0.0/7.0 | 7.9/13.9 |
| AVG | 9.01557 | **9.01602** | 291 | 212 | 3033/9262 | 15057/149 | 29.3 | 7.1 | 0.0/7.0 | 8.0/14.0 |

while iRBBS-0 takes the most nSk and iRBBS-inf takes the least nSk; Compared with iRBBS-inf, iRBBS-0.1 is about 2.7x faster for Dataset D.2. Compared with iRBBS-0, iRBBS-0.1 is about 1.4x faster for Dataset D.2; (iii) compared with R(A)BCD, iRBBS-0.1 can take significantly less nGrad and may take a bit more nSk. This makes it about 7x faster than RABCD and about 12.8x faster than RBCD for Dataset D.2. Besides, for instances D1/D8, D2/D4, D3/D4, D4/D8, RBCD meets the maximum iteration number.

From the above results, we can conclude that iRBBS generally performs much better than R(A)BCD for the two datasets. More importantly, our methods adopt the adaptive stepsize without needing to tune the best stepsize as done in R(A)BCD.

## D.2 Comparison on Computing the PRW Distance (1)

In this subsection, we present more numerical results on Datasets D.1 and D.2 to illustrate the effectiveness and efficiency of our proposed REALM, namely, Algorithm 1. The subproblem (11) is solved by our developed iRBBS algorithm, namely, Algorithm 2, with $\theta_t$ chosen as in Section 4.1.

The results for Dataset D.1 are presented in Table 4. For each $(n, d)$ pair, we randomly generate 10 instances, each equipped with 5 randomly generated initial points. We conside REALM-$(0.055, 0.9)$ and REALM-$(0.02, 0)$ both with $\eta_1 = 1$, $\gamma_\epsilon = 0.25$, and $\gamma_\eta = 0.5$. Note that the latter admits a smaller $\eta_{\min}$ and does not update the multiplier matrix. From Table 4, we can observe that REALM-$(0.055, 0.9)$ can not only return better solutions than REALM-$(0.02, 0.9)$ but also is about 4x faster. This shows that updating the multiplier matrix does help. In fact, on average REALM-$(0.055, 0.9)$ updates the multiplier matrix 8 times in 14 total iterations.

The results over 20 runs on the real Dataset D.2 are reported in Table 5. We consider REALM-$(1, 0)$ and REALM-$(3, 0.9)$ both with $\eta_1 = 200$ and $\gamma_\eta = \gamma_\epsilon = 0.25$. From Table 5, we can see that updating the multiplier matrix also helps. Compared with REALM-$(1, 0)$, REALM-$(3, 0.9)$ can not only return better solutions but is about 2.4x faster. On average, REALM-$(3, 0.9)$ updates the multiplier matrix 6.7 times in 11.7 total iterations.

Table 5: The average results of REALM for Dataset D.2, "a" and "b" stand for REALM-$(1, 0)$ and REALM-$(3, 0.9)$, respectively.

| data | $10^{-3} \times \widehat{\mathcal{P}}_k^2$ | | nGrad | | $\mathrm{nSk_{exp}}/\mathrm{nSk_{log}}$ | | time | | iter | |
|------|------|------|------|------|------|------|------|------|------|------|
| | a | b | a | b | a | b | a | b | a | b |
| D0/D1 | **0.97509** | 0.97508 | 238 | 482 | 770/1406 | 4468/0 | 13.9 | 5.6 | 0.0/5.0 | 8.0/13.0 |
| D0/D2 | **0.79482** | 0.79454 | 340 | 409 | 1473/2636 | 5212/0 | 15.7 | 3.3 | 0.0/5.0 | 3.0/8.0 |
| D0/D3 | 1.20244 | **1.20571** | 359 | 740 | 1758/4162 | 6737/0 | 25.3 | 5.7 | 0.0/5.0 | 8.0/13.0 |
| D0/D4 | 1.22100 | **1.23156** | 428 | 632 | 2169/3599 | 6564/0 | 20.8 | 4.8 | 0.0/5.0 | 8.0/13.0 |
| D0/D5 | **1.03638** | 1.03634 | 539 | 693 | 2385/3840 | 4666/0 | 21.0 | 4.4 | 0.0/5.2 | 8.0/13.0 |
| D0/D6 | 0.80313 | **0.80683** | 313 | 1215 | 1881/2929 | 12402/0 | 17.6 | 8.7 | 0.0/5.1 | 8.0/13.0 |
| D0/D7 | **0.85624** | 0.85624 | 297 | 562 | 928/1907 | 6342/0 | 11.7 | 4.5 | 0.0/5.0 | 8.0/13.0 |
| D0/D8 | 1.05330 | **1.05361** | 292 | 696 | 1575/2091 | 7139/0 | 11.8 | 5.0 | 0.0/5.0 | 7.0/12.0 |
| D0/D9 | 1.08510 | **1.08899** | 306 | 766 | 1780/2386 | 7557/0 | 13.9 | 5.8 | 0.0/5.0 | 8.0/13.0 |
| D1/D2 | **0.66502** | 0.66501 | 193 | 535 | 1042/1068 | 6793/0 | 10.8 | 6.7 | 0.0/5.0 | 8.0/13.0 |
| D1/D3 | 0.86173 | **0.86205** | 484 | 1069 | 1348/2261 | 10720/0 | 22.3 | 12.1 | 0.0/5.7 | 7.6/12.6 |
| D1/D4 | **0.66703** | 0.66702 | 267 | 648 | 860/1372 | 5138/0 | 13.1 | 7.1 | 0.0/5.0 | 8.0/13.0 |
| D1/D5 | **0.83950** | 0.83796 | 639 | 1330 | 1800/2966 | 18003/0 | 19.2 | 10.5 | 0.0/5.1 | 6.0/11.0 |
| D1/D6 | **0.79633** | 0.79614 | 567 | 1243 | 1527/2918 | 12372/0 | 26.9 | 12.8 | 0.0/5.1 | 6.0/11.0 |
| D1/D7 | **0.57276** | 0.57274 | 318 | 862 | 755/2013 | 7149/0 | 19.9 | 10.1 | 0.0/5.0 | 8.0/13.1 |
| D1/D8 | **0.87915** | 0.87737 | 1554 | 1465 | 3318/10079 | 22317/0 | 92.9 | 16.1 | 0.0/5.2 | 7.0/12.0 |
| D1/D9 | **0.85394** | 0.85393 | 326 | 712 | 880/1578 | 5716/0 | 15.8 | 8.3 | 0.0/5.0 | 8.0/13.0 |
| D2/D3 | **0.71916** | 0.71916 | 323 | 858 | 1755/1892 | 16797/0 | 12.3 | 7.5 | 0.0/5.0 | 8.0/13.0 |
| D2/D4 | **1.08616** | 1.08614 | 348 | 853 | 3854/1156 | 7422/0 | 8.3 | 6.5 | 0.0/5.0 | 8.0/13.0 |
| D2/D5 | 1.07706 | **1.08610** | 574 | 1083 | 707/6040 | 7035/1642 | 33.0 | 15.0 | 0.0/5.0 | 8.0/13.0 |
| D2/D6 | **0.90032** | 0.90030 | 342 | 981 | 1022/1339 | 10140/0 | 8.6 | 7.5 | 0.0/5.0 | 6.0/11.0 |
| D2/D7 | 0.69549 | **0.70124** | 272 | 538 | 1890/2284 | 7972/0 | 21.0 | 6.0 | 0.0/5.0 | 7.0/12.0 |
| D2/D8 | **0.67232** | 0.67159 | 336 | 1210 | 1263/2180 | 23641/0 | 13.5 | 10.2 | 0.0/5.0 | 7.0/11.9 |
| D2/D9 | 1.05704 | **1.06966** | 502 | 721 | 3027/4541 | 4172/0 | 28.0 | 5.9 | 0.0/5.0 | 8.0/13.0 |
| D3/D4 | **1.20147** | 1.20145 | 400 | 861 | 1449/1896 | 6847/0 | 11.8 | 6.5 | 0.0/5.0 | 8.0/13.0 |
| D3/D5 | **0.58300** | 0.58299 | 295 | 435 | 1284/1613 | 5758/0 | 8.8 | 3.1 | 0.0/5.0 | 4.0/9.0 |
| D3/D6 | 1.23233 | **1.23234** | 299 | 750 | 1166/1290 | 7664/0 | 7.9 | 5.6 | 0.0/5.0 | 8.0/13.0 |
| D3/D7 | 0.72528 | **0.72529** | 314 | 719 | 836/1263 | 9008/0 | 8.4 | 5.8 | 0.0/5.0 | 6.0/11.0 |
| D3/D8 | **0.88074** | 0.88071 | 334 | 781 | 1433/2175 | 13688/0 | 13.0 | 6.2 | 0.0/5.2 | 6.0/11.0 |
| D3/D9 | 0.83143 | **0.83145** | 363 | 983 | 1552/2159 | 20490/0 | 13.6 | 8.5 | 0.0/5.0 | 8.0/13.0 |
| D4/D5 | 1.00719 | **1.00720** | 302 | 666 | 995/1569 | 8687/0 | 8.3 | 4.5 | 0.0/5.0 | 6.0/11.5 |
| D4/D6 | **0.84488** | 0.84485 | 366 | 650 | 935/1550 | 6296/0 | 9.1 | 4.4 | 0.0/5.0 | 4.5/9.4 |
| D4/D7 | **0.79120** | 0.79065 | 680 | 856 | 2701/3931 | 12068/0 | 24.3 | 6.8 | 0.0/5.2 | 6.0/11.0 |
| D4/D8 | 1.09898 | **1.09899** | 308 | 566 | 1307/1616 | 6729/0 | 9.5 | 4.3 | 0.0/5.0 | 7.0/12.0 |
| D4/D9 | **0.49061** | 0.48902 | 304 | 437 | 1620/1933 | 10152/0 | 11.4 | 3.8 | 0.0/5.0 | 4.0/9.0 |
| D5/D6 | **0.71801** | 0.71769 | 337 | 628 | 1275/1716 | 8743/0 | 8.9 | 4.2 | 0.0/5.0 | 4.0/9.0 |
| D5/D7 | 0.91332 | **0.91333** | 287 | 649 | 744/928 | 5357/0 | 5.6 | 4.4 | 0.0/5.0 | 8.0/13.0 |
| D5/D8 | **0.71623** | 0.71573 | 323 | 823 | 1633/2654 | 17832/0 | 13.8 | 6.0 | 0.0/5.0 | 6.0/11.0 |
| D5/D9 | **0.77656** | 0.77597 | 371 | 811 | 1952/2633 | 14998/0 | 14.4 | 5.9 | 0.0/5.0 | 5.0/10.0 |
| D6/D7 | **1.11056** | 1.11051 | 340 | 969 | 843/790 | 4240/0 | 5.7 | 6.8 | 0.0/5.0 | 8.0/13.0 |
| D6/D8 | 0.91820 | **0.91821** | 267 | 768 | 1258/1703 | 14078/0 | 9.6 | 5.9 | 0.0/5.0 | 8.0/13.0 |
| D6/D9 | **1.10662** | 1.10607 | 368 | 550 | 1029/1712 | 4346/0 | 10.4 | 3.9 | 0.0/5.0 | 3.0/8.0 |
| D7/D8 | **1.07946** | 1.07946 | 298 | 825 | 909/1188 | 6158/0 | 7.6 | 6.2 | 0.0/5.0 | 8.0/13.0 |
| D7/D9 | 0.60736 | **0.60796** | 382 | 606 | 1794/2195 | 11288/0 | 13.8 | 5.2 | 0.0/5.0 | 4.0/9.0 |
| D8/D9 | **0.86860** | 0.86855 | 287 | 595 | 1187/1857 | 8905/0 | 10.9 | 4.6 | 0.0/5.0 | 5.0/10.0 |
| AVG | 0.88606 | **0.88697** | 386 | 783 | 1504/2378 | 9551/36 | 16.1 | 6.7 | 0.0/5.0 | 6.7/11.7 |

