# OpenReview forum: "A Riemannian Exponential Augmented Lagrangian Method for Computing the Projection Robust Wasserstein Distance"
_NeurIPS.cc/2023/Conference — NeurIPS 2023 poster_

### Official Review · Reviewer_EBs3 · 2023-07-04

**Soundness:** 3 good
**Presentation:** 3 good
**Contribution:** 3 good
**Rating:** 5
**Confidence:** 3

**Summary:**

This paper considered the problem of computing the projection robust Wasserstein distance between two discrete probability measures. By formulating this problem as an optimization problem over the product space of the Stiefel manifold and the Euclidean space with additional nonlinear inequality constraints, the authors proposed the so-called Riemannian exponential augmented Lagrangian method (REALM) to solve them. Further, the convergence of REALM was given. For solving the subproblems in REALM, the authors designed the so-called inexact Riemannian Barzilai-Borwein method with Sinkhorn iteration (iRBBS), where stepsizes are adaptively chosen. As the authors claimed, the complexity of iRBBS to attain an $\epsilon$-stationary point of the original PRW distance problem matches the best known iteration complexity result.

**Strengths:**

Clearly, this paper generalized some results of the references [26,32]. To the reviewer's best understanding, the core idea is to find feasible points that satisfy the first-order necessary conditions of problem (6) or (11). To solve a three-block optimization problem, the paper transformed it as alternatively minimizing the Stiefel manifold variable $U$ and the Euclidean variables $\alpha$ and $\beta$, where the later can be solved by the well-established inexact gradient methods. Overall, this paper is well-written and mathematically solid.

**Weaknesses:**

(1) Some mathematical details/arguments are missing (please point out if they are added in the supplementary material). For examples, Line 115: why the minimizer $(x^*,y^*)$ of the problem (9) must satisfy the relationship $y^*=\eta\log(\Vert\zeta_\eta(x^*,11^T)\Vert_1)$? Line 124, how to calculate the gradient of $\mathcal{L}_{\eta_k}(x,\pi^k)$?

(2) It seems that the update of $\theta_{t}$ in Algorithm 2 is missing because $\theta_{t+1}$ appears in the inexactness criterion (19b). Please correct me if not.

(3) The paper may require some ablation studies in the numerical experiments. The authors claimed REALM always outperforms the Riemannian exponential penalty approach since it could avoid too small penalty parameters in many cases (Lines 70-71). What's the range for ``too small" penalty parameters? And which cases?

**Questions:**

1. Line 65: One ``method" in this line is redundant.

2. Line 91: There is no definition for $\nabla f(U)$. The metrics on the left and on the right side are different.

3. Line 145, Proposition 2.6: How to come out with the $x^s$? What is the motivation for the definition of $x^s$?

4. Line 252: Should be $\theta_t$ in the place of $\theta$?

**Limitations:**

Yes

---

> ### Author Rebuttal · Authors · 2023-08-08
>
> Many thanks for your positive comments on the basic results achieved in the paper and your helpful comments and suggestions. The following are our point-to-point responses to your comments.
> - Response to Weakness 1.
> Thank you for your comments. We acknowledge that, due to the page limit, some mathematical details/arguments are missing in the paper. We shall add them either in the revised paper or in the supplementary material. Below let us first provide the detailed derivations for the two examples you mentioned in your above comments.
>    - Due to the optimality of $(\\mathbf{x}^*, y^*)$, we have $\nabla\_{y} \widetilde{\mathcal{L}}\_{\eta}(\mathbf{x}^*, y^*, 11^T) =0.$ Recalling that  $\widetilde{\mathcal{L}}\_{\eta}(\mathbf{x}^*, y^*, 11^T) = r^T \alpha + c^T \beta + y + \eta \sum_{ij} \exp\left (- \frac{\varphi(\mathbf{x})\_{ij} + y}{\eta} \right),$ we obtain $\nabla\_{y}\widetilde{\mathcal{L}}\_{\eta}(\mathbf{x}^*, y^*, 11^T) = 1  - \sum\_{ij} \exp\left (- \frac{\varphi(\mathbf{x^*})\_{ij} + y^*}{\eta} \right) = 0$,  which gives  $y^* =  \eta \log \sum\_{ij} \exp\left (-\frac{\varphi(\mathbf{x^*})\_{ij}}{\eta} \right)   =  \eta \log (\\|\zeta\_{\eta}(x^*, 11^T)\\|\_1).$
>     - By the expression
> \\[\mathcal{L}\_{\eta}(\mathbf{x}, \pi) = r^T \alpha + c^T \beta + \eta \log \sum\_{ij} \pi\_{ij} \exp\left( -\frac{\alpha\_i + \beta\_j + \langle M\_{ij}, UU^T\rangle} {\eta}\right)
> \\]
> and employing the chain rule,  we can compute the gradient of $\mathcal{L}\_{\eta_k}(\mathbf{x}, \pi^k)$ with respect to $U$  as
> \\[\nabla\_U \mathcal{L}\_{\eta\_k}(\mathbf{x}, \pi^k) = - 2  V\_{\phi\_{\eta\_k}(\mathbf{x}, \pi^k)} U\\] with
> $V\_{\phi_{\eta\_k}(\mathbf{x}, \pi^k)}  = \sum\_{ij} [\phi\_{\eta\_k}(\mathbf{x}, \pi^k)]\_{ij} M\_{ij}$ (and $V_{\pi}$ for a given matrix $\pi$ is defined in Line 85 of the paper). Moreover, the gradient of $\mathcal{L}_{\eta_k}(\mathbf{x}, \pi^k)$ with respect to $\alpha$ and $\beta$ can be found in Lines 124 and 125, respectively.
>
> - Response to Weakness 2.
> Thanks for this important comment. Theoretically, Algorithm 2 allows $\theta_t$ to be any nonnegative number, which is mainly due to the nice property shown in Lemmas B.1 and B.2 of the Sinkhorn iteration. Please see also Theorem B.5 for the convergence result of Algorithm 2. However, it would be better to clearly specify the choice/update of $\theta_t$ in Algorithm 2, as you suggested. In the revision, we shall add the update of $\theta_t$, namely, $$\theta_{t+1} = \max\\{\theta/(2\\|C\\|_\infty) \\|\xi^t\\|_F, \epsilon_2\\},$$  in Line 6 of Algorithm 2 and include the phrase ``Set  $\theta_0 = 1$ and choose $\theta > 0$'' in its input part.  The response also relates to your Question 4 below.
>
> - Response to Weakness 3.
> Thanks for your important comment. There have already been systematic studies in the literature on the advantage of the exponential ALM over the exponential penalty approach in terms of better numerical stability due to its ability of avoiding too small penalty parameters; please check Refs. [18], [45] and [50] in the paper for your reference. However, it is hard to give an explicit estimate on the range ``too small''.  Based on our numerical experience, we have found that when $\eta_k\leq 1/500\max\\{||r||\_\infty,||c||\_\infty\\}$ or $\eta_k \leq 1/900 \\|C(U^t) - \eta_k \log \pi^k\\|_{\mathrm{var}}$ (as shown in Line 283 and Line 284, respectively), the penalty parameter would be considered too small (when REALM is used to solve the PRW distance problem). In such cases, we recommend performing the Sinkhorn iteration with log-domain stabilization, as shown in Eq. (21) and Line 301.
> - Response to Question 1.
> Thanks. We shall correct this typo in the revision.
> - Response to  Question 2.
> Thank you for bringing this to our attention. You are correct, and we apologize for the oversight in not making this point clear. The notation $\nabla f(U)$ means the Euclidean gradient of $f$ with respect to $U$. Usually, the metrics on the two sides of $\langle \mathrm{grad} f(U), \xi \rangle = \langle \nabla f(U), \xi \rangle$ are different: the left-hand side should be $\langle \cdot, \cdot \rangle_U$ which is a smoothly varied inner product on the tangent space at $U$.  Since the Stiefel manifold is an embedded manifold of a Euclidean space $\mathcal{E}$, we can simply choose $\langle \cdot, \cdot \rangle_U$ as the standard inner product in $\mathcal{E}$, which coincides with the metric in the right-hand side.  We will clarify this issue in the revision.
> - Response to Question 3.
> Thanks for this important comment. It comes out with an observation that
> \begin{align}
> &r^T (\alpha + \upsilon_1 \mathbf{1}) + c^T (\beta + \upsilon_2 \mathbf{1}) + \eta \log \sum_{ij} \pi_{ij} \exp\left( -\frac{(\alpha_i + \upsilon_1) + (\beta_j + \upsilon_2) + \langle M_{ij}, UU^T\rangle} {\eta}\right)  \\\\
> ={}& r^T \alpha   + c^T \beta  + \eta \log \sum_{ij} \pi_{ij} \exp\left( -\frac{\alpha_i  + \beta_j  + \langle M_{ij}, UU^T\rangle} {\eta}\right),
> \end{align}
> where the equality uses the fact that $r^T \mathbf{1} = c^T \mathbf{1} = 1$.  Hence, by carefully choosing $\upsilon_1$ and $\upsilon_2$, we can derive $\mathbf{x}^s$ and the corresponding results in Proposition 2.6.
> Moreover, the first two requirements in Eq. (14) are essential to establish the boundness of $\alpha^k$ and $\beta^k$, which in turn ensures the convergence of REALM.  This is the main motivation behind defining such $\mathbf{x}^s$.
> - Response to Question 4.
> Thank you for your comment. The parameter $\theta$ is a pre-selected constant in the proposed iRBBS algorithm. By choosing different values of $\theta$, we obtain different versions of iRBBS, denoted as iRBBS-$\theta$. Usually, a smaller $\theta$ means that the subproblem is solved more accurately. We will clarify the meaning of $\theta$ in Line 252 to avoid any potential confusion. This clarification will be included in the revision.

---

> > ### Comment · Reviewer_EBs3 · 2023-08-17
> >
> > Thank you for your rebuttal. I have now understood your detailed explanations. I would like to keep my recommendation.

---

### Official Review · Reviewer_iBqS · 2023-07-06

**Soundness:** 3 good
**Presentation:** 3 good
**Contribution:** 3 good
**Rating:** 5
**Confidence:** 3

**Summary:**

This paper reformulates the projection robust Wasserstein distance as an optimization problem over the product of the Stiefel manifolds and a subset of a Euclidean space. A Riemannian exponential augmented Lagrangian method (REALM) is proposed to solve this problem. The proposed method is empirically more stable than the existing Riemannian exponential penalty-based approach. A Riemannian BB method with Sinkhorn iteration (iRBBS) is used for the subproblem. The iteration complexity of iRBBS is given. Numerically, the proposed method outperforms the existing methods.

**Strengths:**

(1) A method (REALM) is proposed and its global convergence is given.
(2) An inexact Riemannian BB method with Sinkhorn is developed and its iteration complexity is derived. Such a result is interesting by itself.
(3) From the numerical comparison, the proposed method is most efficient.


**Weaknesses:**

The existing methods have iteration complexities for the corresponding algorithms. This paper only gives the iteration complexity for the subproblem.


**Questions:**

(1) Can the authors give the iteration complexity for the overall algorithm? What is the main difficulty here?
(2) What is the dominated computational time in the proposed algorithm and the compared algorithms?


**Limitations:**

The authors pointed out an important limitation in the theoretical analysis, which is the lack of a lower bound of \eta_k.

---

> ### Author Rebuttal · Authors · 2023-08-08
>
> - Response to Question 1.
> Many thanks for this insightful comment. Regarding the current overall Algorithm 1,  we were not able to establish the iteration complexity due to the following two main difficulties: (i) characterizing the connection between the two complementarity measures $\\|W^k\\|\_F$ and $\langle \pi^k, Z(\mathbf{x}^k)\rangle$ at the approximate stationary point of the subproblem; and (ii) establishing the relationship between $\eta_k \tilde{\pi}^{k+1}\_{ij}$ and $\varphi(\mathbf{x}^k)\_{ij}$.
>
>    However, thanks to Theorem 2.4, we can now slightly modify Algorithm 1 to establish the iteration complexity. More specifically, let  $e^k = \langle \tilde \pi^{k}, Z(\mathbf{x}^k)\rangle$, where $\tilde \pi^{k}: = \mathsf{Round}(\phi_{\eta_k}(\mathbf{x}^k, \pi^k), \Pi(r,c))$ is a feasible matrix returned by the rounding procedure mentioned in Theorem 2.4. By modifying the ``if'' condition in Line 6 of Algorithm 1 as
>    \\[
>    \\|W^{k}\\|\_F \leq \gamma\_W \\|W^{k-1}\\|\_F \\quad \\mathrm{and}\\quad  e^k \leq \gamma_W e^{k-1},
>    \\]
>    we can establish the iteration complexity of the whole exponential ALM without compromising its global convergence. By leveraging the connection between the approximate stationary points of the subproblem and the original problem as proven in Theorem 2.4, we can ensure that the algorithm will terminate within at most
>    \\[
>    \\mathcal{O}\\left(\\max\\left\\{\\log \\frac{1}{\epsilon\_1}, \\log \\frac{1}{\epsilon\_2}, T\_k\\right\\}\\right)
>    \\] iterations, where $T_k := \min \\{k \mid \varrho_k \leq \epsilon_c\\}$.  We will incorporate these improved results in the revised version to address your concerns. Thank you once again for your valuable comment.
>
> - Response to Question 2.
> Thanks very much for this important comment. The main computational cost in the proposed iRBBS and R(A)BCD proposed by (Huang et al. 2021a) lies in the following steps:
>    - Compute the inexact Riemannian gradient $\xi^t$, whose cost is $\mathcal{O}(ndk + n^2k + dk^2)$.
>    - Update  $U^{t+1} = \mathrm{Retr}_{U^t}(-\tau_t \xi^t)$, whose cost is $\mathcal{O}(dk^2)$.
>    - Compute the matrix $A \in \mathbb{R}^{n \times n}$ with $A_{ij} = \pi_{ij}^k \exp(-\langle M_{ij}, UU^T\rangle/\eta_k)$ (shown in Line 255),  whose cost is $\mathcal{O}(n^2)$.
>    - Perform the Sinkhorn iteration (21) or (27),  whose cost is $\mathcal{O}(n^2)$.
>
>    Note that the cost of verifying whether the linesearch condition (25) holds is low since $\mathcal{L}(\mathbf{x}^{t+1})$ is always equal to $r^T \alpha^{t+1} + c^T \beta^{t+1}$ due to $\\|\zeta^{(\ell)}\\|_1 = 1$, as shown below Eq. (21).
>
> - Response to Limitations.
> Thanks very much for pointing out this important issue.  Now we can overcome this limitation by assuming the Riemannian versions of the following three conditions, including the linear independence constraint qualification, the strict complementarity condition,  and the second-order sufficient condition.  This extension builds upon the results of Echebest et al. (2016) (listed as Ref. [18] in the paper). We will incorporate this result and highlight the significance of considering these Riemannian conditions in the revision.

---

> > ### Comment · Reviewer_iBqS · 2023-08-17
> >
> > Thanks for the clear discussions. I increased the score to 5.

---

### Official Review · Reviewer_X3cc · 2023-07-07

**Soundness:** 3 good
**Presentation:** 2 fair
**Contribution:** 3 good
**Rating:** 6
**Confidence:** 1

**Summary:**

The authors first reformulate the computation of the PRW distance as an optimization problem over the Cartesian product of the Stiefel manifold and the Euclidean space with additional nonlinear inequality constraints. And then they also propose a Riemannian exponential augmented Lagrangian method (REALM) for solving the problem


**Strengths:**

1.	The authors propose a Riemannian exponential augmented Lagrangian method (REALM) method to efficiently and faithfully compute the PRW distance and establish the global convergence of REALM in the sense that any limit point of the sequence generated by the algorithm is a stationary point of the original problem

2.	To efficiently solve the subproblem in REALM, the authors also propose a novel and practical algorithm, namely, the inexact Riemannian Barzilai-Borwein (BB) method with Sinkhorn iteration (iRBBS).


**Weaknesses:**

1.	The authors should provide a proof sketch for their main theoretical analysis in this paper.

2.	The authors should add more experimental results to verify their theoretical results and their algorithms.


**Questions:**

See the above setion

**Limitations:**

See Weaknesses

---

> ### Author Rebuttal · Authors · 2023-08-08
>
> - Response to Question 1.
> Many thanks for your suggestion. We agree with you that it is always helpful to provide a proof outline or a preview of the proof before delving into detailed theoretical analysis. We shall take your suggestion into account when we revise the paper.
>
> - Response to Question 2.
> Thanks for this comment. We shall try to provide more simulation results to better verify the obtained theoretical results.

---

> > ### Comment · Reviewer_X3cc · 2023-08-20
> >
> > Thank you for your rebuttal. The rebuttal has clarified my questions and I decided to keep my score.

---

> ### Comment · Area_Chair_4nza · 2023-08-19
>
> Dear Reviewer X3cc: can you take a look at the authors' rebuttal, and see if your comments are addressed?

---

### Official Review · Reviewer_KEsx · 2023-07-21

**Soundness:** 3 good
**Presentation:** 2 fair
**Contribution:** 2 fair
**Rating:** 5
**Confidence:** 1

**Summary:**

This work proposes a new method, called REALM, to compute the projection robust Wasserstein (PRW) distance.

The method REALM is an extension of the exponential augmented Lagrangian method to the Riemannian space.

The convergence of REALM is established.

To solve a subproblem during REALM, this work proposes an inexact Riemannian Barzilai-Borwein method with Sinkhorn iteration (iRBBS).

The complexity rate of iRBBS to solve the subproblem, whose solution is an $(\epsilon_1,\epsilon_2)$-stationary point of the PRW problem, matches the existing works in the literature.

This work claims the robustness in terms of the parameter-tuning in their proposed algorithms.

**Strengths:**

This work proposes a new method to compute the PRW distance with solid theoretical guarantees.

**Weaknesses:**

The complexity rate matches the existing rate. The idea of reformulating the PRW distance computing problem into a Riemannian optimization setting is not new, e.g. Huang et al. 2021. The claimed easier parameter-tuning part seems to need more elaboration, either from the perspective of theory, or from numerical evidences.

**Questions:**

Hello authors, I have the following questions,

1. Generally speaking, the augmented Lagrangian method also suffers from the poor choice of penalty parameters, e.g., see Curtis et al. 2015. In your work, it claims that REALM can "potentially avoid too small penalty parameters". I was wondering if you could provide some intuition here (to explain why) or explicitly point out any efforts on the algorithmic design to have this property.

2. When you extend the exponential ALM, you claim it is a nontrivial extension because the specific problem structure encourages the specific conditions (14) and (16). Besides this, is there anything significantly nontrivial?

3. The iRBBS is discussed in Part 3. In particular, you mention the cost of updating $\alpha$ and $\beta$ is much less than that of updating $U$. This feature appears to be a valuable addition to your framework (17). Could you confirm my understanding? Is it possible to elaborate/support your proposed iRBBS, e.g., with respect to its implementation simplicity, to make your work more distinctive from existing literature.


**Limitations:**

In the Euclidean space, some existing works consider the boundedness of the penalty parameters, e.g., see Echebest et al. 2015. While the additional (Riemannian, inequality) constraints might make situation totally different, I was wondering if those existing literature could shed light on further investigating the parameters settings.

---

> ### Author Rebuttal · Authors · 2023-08-08
>
> Many thanks for your positive comments on the basic results achieved in this paper and your helpful comments and suggestions.
> - Response to Weakness.
> Thanks for your comments.  We want to take this opportunity to elaborate more on the nonmonotone linesearch condition (25) to clarify this issue from the theory perspective.
> In particular, the interested problem (11) is a three-block optimization problem and Huang et al. 2021 proposed to solve it by R(A)BCD. The current paper treats it as a one-block optimization problem with $U$ as the only variable, as shown in problem (17). The new perspective allow applying the (inexact) gradient algorithm to solve problem (17); see the proposed iRBBS in Algorithm 2. The key difference here is that the stepsizes in iRBBS can be adaptively chosen with the help of the linesearch condition (25), but tuning the stepsizes for updating $U$ is not easy for R(A)BCD. This is one of the two main contributions of the current paper. The other benefits of treating the interested problem as a one-block optimization problem as in (17) can be found in our responses to your Question 3.
> - Response to Question 1.
> Thanks for your important comment. The intuition in using the proposed exponential ALM lies in its equivalence to a proximal point method for solving the dual problem, incorporating an entropy function. The estimate of the multiplier matrix $\pi^k$  can be viewed as the proximal point center, and a varying center $\pi^k$ typically outperforms a fixed center $\pi^k \equiv 11^T$ in the proximal point method.
> To further improve the numerical performance (in particular the numerical stability) of the proposed REALM, we have carefully exploited the structure of the PRW distance problem and used the problem's special structure into the general exponential ALM framework. More specifically, the first algorithmic innovation is the conditions in (14), which are crucial to guarantee the boundness of the iterates. The second algorithmic innovation is the simple rules of updating the penalty parameters and the multiplier matrices, which take into account of the specific connections between approximate stationary points of the subproblem and the original problem. The latter is of great importance to both the global convergence and the numerical performance/stability of the proposed REALM.
>
> - Response to Question 2.
> Thank you for this important comment, which gives us an opportunity to clarify this point. In addition to the two specific conditions (14) and (16), the following two differences distinguish our proposed REALM from the existing exponential ALM (e.g., the one proposed in Echebest et al. 2016):
>    - **Measure of complementarity.** The measure of complementarity used in our proposed REALM differs from that employed in Echebest et al. 2016. More specifically, the measure in our REALM is motivated by the direct use of the complementarity condition adopted in the classical (quadratic) ALM, while the measure of complementarity used in Echebest et al. 2016 can be regarded as ``an appropriate measure for the exponential case" as mentioned in the second paragraph on Page 96 of their paper.
>    - **Conditions on global convergence.** To guarantee the global convergence of the exponential ALMs, some (strong) constraint qualifications and the boundness of the iterates generally need to be assumed; see Proposition 2.1 and Theorem 2.1 in  Echebest et al. 2016 for the corresponding results. The current paper extends the exponential ALMs to solve a class of inequality-constrained nonlinear optimization problems with manifold constraints. Based on our best knowledge, this is the first Riemannian version of the exponential ALM. In particular, for our considered PRW distance problem,  we can prove the boundness of the iterates generated by the proposed REALM without making the assumption and establish the global convergence of the proposed REALM without explicitly dealing with the constraint qualification assumption. This advantage is mainly due to the aforementioned *essential* changes in the proposed REALM (compared with the existing methods), i.e., specific conditions (14) and (16) on the solution of subproblems and the adopted measure of complementarity.
> - Response to Question 3.
> Your understanding is absolutely correct, and we truly appreciate your valuable suggestion.
> One motivation of proposing iRBBS is that the condition number of the one-block optimization problem could be smaller than that of the three-block optimization problem. Consequently,  employing an inexact gradient descent method to solve the one-block optimization problem would be more efficient than simply using the block coordinate descent approach for solving the corresponding three-block optimization problem. Moreover, the inexact gradient method also provides important and valuable insights into adaptively choosing the stepsize for updating $U$ via the nonmontone linesearch. Last but not least, the fact that updating $\alpha$ and $\beta$ is much easier than updating $U$ also contributes to the superior performance of iRBBS compared to existing block coordinate descent approaches in practice, as you correctly pointed out. Otherwise, there is no necessity to update $\alpha$ and $\beta$ multiple times while only updating $U$ once in each iteration of the proposed iRBBS.
> - Response to Limitations.
> Thank you very much for this insightful comment. After a careful and thorough investigation, we can now establish the boundness of the penalty parameter for the proposed REALM by leveraging the analysis in Echebest et al. 2016. To achieve this, we need to consider the Riemannian versions of the following three conditions: the linear independence constraint qualification, the strict complementarity condition, and the second-order sufficient condition. However, it remains unclear under which (easily checkable) conditions the above three conditions hold true, which requires further investigation.

---

> > ### Comment · Reviewer_KEsx · 2023-08-21
> >
> > Thank you for the clarification, which addresses most of my concerns.

---

> ### Comment · Area_Chair_4nza · 2023-08-19
>
> Dear Reviewer KEsx: can you take a look at the authors' rebuttal, and see if your comments are addressed?

---

### Official Review · Reviewer_g8K2 · 2023-07-31

**Soundness:** 2 fair
**Presentation:** 2 fair
**Contribution:** 2 fair
**Rating:** 5
**Confidence:** 4

**Summary:**

this paper proposed a Riemannian Exponential Augmented Lagrangian Method for solving the projection robust wasserstein distance problem. the authors claimed two contributions compared with the previous works: 1) the proposed algorithm is much more stable as \eta needs to be small in previous works. 2) a Riemannian Barzilai-Borwein method was proposed to adaptively fine tune the step size. numerical experiments compared with previous works were reported.

**Strengths:**

1. the author proposed to do multiple sinkhorn steps and one riemannian gradient step in each iteration because sinkhorn steps are much more cheaper. this idea makes sense and the author provided convergence guarantee for the proposed algorithm.
2. numerical experiments show the advantages of the proposed algorithm compared with RBCD.

**Weaknesses:**

1. the main concern is that the author claimed that the proposed algorithm is numerically more stable than RGAS/RBCD because both of them require \eta to be very small. however, I'm not convinced by simply saying "Based on the knowledge that the exponential ALM is usually more stable than the exponential penalty approach ... ". are there any systematic study to support this claim? the proposed ALM requires the penalty parameter \eta to be exponentially decreasing. such an \eta appears in the denominator of an exponential term (e.g. eq 7). wouldn't this cause the same numerically instability issue? besides, in the proposed algorithm, the author applies the sinkhorn iteration, which will naturally introduces the numerical issues?

2. the writing of this paper needs to be further improved.

**Questions:**

see the weakness

**Limitations:**

see the weakness.

---

> ### Author Rebuttal · Authors · 2023-08-08
>
> Many thanks for pointing out the weakness, which gives us an opportunity to clarify it.
>
> First, there are indeed some systematic studies comparing exponential ALM and exponential penalty approaches. We wish to highlight several main results from the existing literature on this matter:
> - Convex case.
>    - Tseng and Bertsekas [TB93], listed as Ref. [45] in our paper, proved that the penalty parameter $\eta_k$ in the exponential ALM (with subproblems solved exactly) can be chosen as any positive number. This is in sharp contrast to the exponential penalty approach, where the penalty parameter is typically required to be very small or go to zero. Please refer to Proposition 3.1 in [TB93].  Additionally, [TB93]  established the linear convergence rate of the exponential ALM for solving the linear programming; see Proposition 4.1 therein.
>    - More recently, Yang and Toh [YT22], listed as Ref. [50] in our paper, proposed an inexact version of the Bregman proximal point algorithm, which is equivalent to the exponential ALM with subproblems solved inexactly. They proved that the penalty parameter can be chosen as any positive number. Moreover, they claimed that their proposed approach exhibits greater stability than the exponential penalty approach (when applied to solve the standard optimal transport problem). In particular, they mentioned in the paper:  ``*our iBPPA with the entropic proximal term can bypass some numerical instability issues that often plague the popular Sinkhorn's algorithm used in the OT community. This is because in contrast to Sinkhorn's algorithm, our iBPPA does not require the proximal parameter to be very small in order to obtain an accurate approximate solution, as evident from our numerical results*"
> - General nonlinear case.
>    - Under the regular condition, the strict complementarity condition, and the second-order sufficient condition, Dussault [Dus04] demonstrated that the exponential ALM exhibits one-step superlinear convergence with a rate of $4/3$ (see Theorems 1.1 and 2.1 therein). In contrast, the exponential penalty approach takes two-step superlinear convergence with the same rate (see the second paragraph on Page 475). Additionally, Dussault [Dus04] provided a detailed example illustrating that the exponential ALM can be more stable than the exponential penalty approach. For further information and specific results, please refer to Tables 3.1-3.3 in the referenced paper.
>    - Recently, Echebest et al. [ESS16], listed as Ref. [18] in our paper, showed that the penalty parameter $\eta_k$ *can be bounded away from zero* under the conditions of the linear independence constraint qualification, the strict complementarity condition, and the second-order sufficient condition. However, no evidence indicating that the exponential penalty approach enjoys a similar property under these conditions.
>
> Based on the aforementioned studies on the exponential ALM, we stated in our paper ``Based on the knowledge that the exponential ALM is usually more stable than the exponential penalty approach...". However, we apologize for not providing more detailed explanations and justifications in the current version. We will certainly include the corresponding discussion in the revised version.
>
> Second, we acknowledge that the exponential ALM might face a similar numerical instability issue as the exponential penalty approach if the penalty parameter $\eta_k$ becomes too small. However, we hope the following explanations could convince you that the exponential ALM is potentially more stable than the exponential penalty approach for the PRW problem.
> - To address the potential numerical instability caused by the small penalty parameter $\eta_k$, we propose to update $\eta_{k+1}$ as $\eta_{k+1} = \max\\{\gamma_{\eta} \eta_k, \eta_{\min}\\}$ as in Line 279.  This update scheme ensures that the penalty parameter does not become too small during the iterations. We observed from our numerical experiments in Table 2 and Tables 4-5 that, compared to the exponential penalty approach, the exponential ALM can accommodate larger values of $\eta_{\min}$. This advantage is primarily attributed to the update of the multiplier matrices. By allowing for a larger $\eta_{\min}$, the exponential ALM is less likely to suffer from the numerical instability due to the use of a small penalty parameter, as observed in the exponential penalty approach.
> - Theoretically, we can extend the analysis in [ESS16] to prove that the penalty parameter $\eta_k$ in the proposed Riemannian exponential ALM will also be bounded away from zero if the Riemannian versions of the three conditions hold, including the linear independence constraint qualification, the strict complementarity condition, and the second-order sufficient condition. We will include additional remarks in the revision to provide further clarity on this point.  It should be mentioned that these three conditions might not be easy to check since we do not have prior knowledge of the solution.  However, based on our numerical results, we believe that for certain instances, the three conditions indeed hold.
>
> Third, we shall further enhance the clarity of our writing in the revision to ensure that the results are presented more clearly and the paper becomes easier to follow.
>
>  The corresponding references are listed in order.
> - [Dus04]  J.-P. Dussault. Augmented non-quadratic penalty algorithms. *Math. Program.*, 99(3):467-486, 2004.
> - [ESS16]  N. Echebest, M. D. Sánchez, and M. L. Schuverdt.  Convergence results of an augmented Lagrangian method using the exponential penalty function. *J. Optim. Theory Appl.*, 168:92-108,2016.
> - [TB93] P. Tseng and D. P. Bertsekas. On the convergence of the exponential multiplier method for convex programming. *Math. Program.*, 60(1-3):1-19, 1993.
> - [YT22]  L. Yang and K.-C. Toh.  Bregman proximal point algorithm revisited: A new inexact version and its inertial variant. *SIAM J. Optim.*, 32(3):1523-1554, 2022.

---

> > ### Comment · Reviewer_g8K2 · 2023-08-15
> > **reply to authors' rebuttal**
> >
> > thanks for the clarification. I've increased my score to 5

---

### Decision · Program_Chairs · 2023-09-21

**Decision:**

Accept (poster)

**Comment:**

This paper considers the projection robust optimal transport problem. A new algorithm, i.e., Riemannian counterpart of the exponential multiplier method, is proposed to solve this problem. The authors established the convergence analysis and demonstrated the efficiency of the proposed algorithm via numerical experiments. All reviewers are favorable to the acceptance of this paper.